# Unlocking the potential of biogas systems for energy production and climate solutions in rural communities

Tao Luo[1], Bo Shen [2] ✉, Zili Mei[1], Anders Hove[3] & Keyi Ju[4]

On-site conversion of organic waste into biogas to satisfy consumer energy demand has the potential to realize energy equality and mitigate climate change reliably. However, existing methods ignore either real-time full supply or methane escape when supply and demand are mismatched. Here, we show an improved design of community biogas production and distribution system to overcome these and achieve full co-benefits in developing economies. We take five existing systems as empirical examples. Mechanisms of synergistic adjusting out-of-step biogas flow rates on both the plant-side and user-side are defined to obtain consumption-to-production ratios of close to 1, such that biogas demand of rural inhabitants can be met. Furthermore, carbon mitigation and its viability under universal prevailing climates are illustrated. Coupled with manure management optimization, Chinese national deployment of the proposed system would contribute a 3.77% reduction towards meeting its global 1.5 °C target. Additionally, fulfilling others' energy demands has considerable decarbonization potential.

Under many scenarios, fossil fuels will be the dominant energy source until 2050, due to the lack of reliable and convenient renewable energy supply systems[1], whose growing use is causing increased carbon emissions[2]. To alter this trend, biogas (55–65% methane content) utilization, based on the waste-to-energy-produced pattern[3], is considered a viable negative emission path for methane control and fossil fuel replacement. Meanwhile, it also has the potential to deliver the same energy-return-on-investment ratio as fossil gas when considering the fossil energy's ecological cost[1]. To achieve the dual goals of improving clean energy accessibility and addressing climate change, an efficient biogas production and utilization system requires widespread application[4], to substantially increase biogas usage while reducing methane emissions from the identified sources of organic matter management[3], such as treating manure in anaerobic lagoons.

Despite biogas having an extensive history in cooking, heating, and power generation, including its use in biogas-based natural gas production[5], its contribution to the current energy mix and its climate benefits remain modest in rural developing areas[6]. This is largely due to the problem of intermittent renewable energy supply and the incomplete utilization of collected biogas[7]. Taking China as an example, approximately 800 million people, most of whom are living in rural areas, have no access to natural gas or biogas[8]. For now, extensive development of natural gas facilities is not a feasible option, because it would cause serious shortages during peak periods and significantly threaten energy security[9], such as the frequent gas shortages from the coal-to-gas policies implemented in northern China. To alter it, biogas is therefore considered a feasible and essential technology to produce sustainable energy to cover the shortage, as the methane production potential of available manure and crop straw is 73.6 billion $m^3$ $yr^{-1}$ in China[10], which is sufficient to cover both urban/rural household demands in the country[11]. Furthermore, high-quality biogas systems have been recognized as the most efficient strategy to reduce methane emission of organic waste treatment[12]. With manure treatment alone, it could reach $6.4 \times 10^7$ tons $CO_2$-eq $yr^{-1}$ in China[13].

In these circumstances, what is the most efficient biogas system and how we deploy it in developing areas have become hot topics[14]. In the European Union, biogas is mainly utilized in either a Combined Heat and Power (CHP) unit or a biomethane upgrading system, and the

[1]Biogas Institute of Ministry of Agriculture and Rural Affairs, Chengdu, China. [2]Lawrence Berkeley National Laboratory, Berkeley, CA, USA. [3]The Oxford Institute for Energy Studies, Oxford, UK. [4]Jiangsu University of Science and Technology, Zhenjiang, China. ✉e-mail: boshen@lbl.gov

generated energy is transmitted to the power grid or gas distribution network in real time and consumed in the form of guaranteed purchases[15]. These running models require high quality monitoring devices and automatically controlled processes to ensure their performance[16]. Their wide use in developing areas would face various challenges of economic feasibility and high level operational requirements, owing to both inadequate subsidies and insufficient technical support[17]. Practically, on-site generation and direct supply biogas to consumers would result in significant advantages to the supply chain of exploitation, conversion, and distribution[18]. Most of all, it is possible to acquire the best co-benefits of sustainable energy production and greenhouse gas (GHG) emission mitigation[19], on the prerequisites of on-demand biogas supply and close-to-zero methane leakage.

A community biogas production and distribution system (CBPD), with high availability of organic matter and direct generation of clean energy to alleviate poverty in rural areas, is deemed the most feasible choice to provide an adjustable, timely, and flexible biogas supply with the largest range of applications[20]. To maximize the CBPD's co-benefits potential, the amount of biogas consumed on the user side should be equal or close to that of biogas production on the plant side, which represents a consumption-to-production ratio (CPR) of 1. A higher CPR is more helpful for climate change mitigation, which means a greater percentage of collected methane utilized, and less methane discharged into the open air[7]. In this case, a proposed CBPD with CPR of close to 1 would have the dominant advantages over conventional energy supply systems in developing economies[21], as the adequacy of feedstock for biogas conversion and customers for biogas usage in rural area are the present favorable conditions.

In this study, exploring a CBPD optimization framework and its operational strategies are keys to forming the self-sufficient system, which would be capable of independent operation and could provide innovative solutions distinguishing it from those found in conventional systems[22]. Thus, we report an upgraded CBPD with enhanced features to maximum the co-benefits of sustainable energy production and climate solutions for broad application, based on Chinese case studies; and provide a quantitative demonstration to analyze the dynamic changes of biogas flow depending on the user side and production side. Its general form includes fitting the biogas supply rate curve based on the characteristics of user-side consumption, establishing a self-adjusting platform, and building the mechanism of biogas flow modulation. We also show that an upgraded CBPD is a viable method for realizing the full co-benefits of transforming organic waste into biogas under prevailing climate conditions. Widely deploying the proposed CPBD in China can make an important contribution toward meeting its global 1.5 °C target.

## Results
### Optimization of CBPD with demand-dependent biogas production
The proposed general form of self-adjusting biogas flow is shown in Fig. 1a. The optimization was categorized into four analytical steps: a data-driven identification of the biogas demand rate; quantifying operational parameters of biogas production; temporal biogas storage capacity designs to avoid discontinuous supply and biogas loses; and operational strategy determination for coordinating biogas flows both on the plant side and user side (Fig. 1b).

In the first step, the curve of biogas consumption rate was simulated to conceptualize corresponding curves of production rates and their requirement of buffer capacities. The status-quo characteristics were identified based on the communities' use data analyses, incorporating both routine activities (such as cooking) and intermittent activities (such as heating in cold weather). Furthermore, there were some potential biogas customers, included households, restaurants, local agriculture factors, and so on. Once they completely understood the characteristics of energy supply on demand, i.e., biogas is good for

the environment[23], and free of a carbon tax, overall biogas consumption could be predicted to increase with time, both temporally and spatially. Thus, forecasting these multiple conditions in advance was definitely essential to design a reliable production and flexible supply system.

In the second step, the biogas production capacity was quantified according to the common knowledge of fermentation temperature and organic load rate[24]. The raw biogas production curve could be determined using variables related to the reactor configuration and operational parameters, and existing mathematical models or improved equations derived from related operational data fitting[7]. Sensitivity analyses and refinement of final biogas production curve, modified for amplitude and duration, would be carried out according to the actual data-driven modeling and step 4, because feeding intervals and feeding time points could play essential roles in pairing the biogas flows on the plant side and user side[25].

In the third step, biogas storage capacity was determined to efficiently store excess output of biogas for dynamically meeting biogas production shortages, which was based on precise estimates of biogas consumption rate, accurate feeding process controls, and efficient margin designs. Among these, margin design is safeguards to achieve the target that CPR is equal to 1 on site, mainly determined by the operation levels of feeding, deviations in fermentation temperatures, irregular temporal and quantitative changes of biogas usage, and so on.

In the fourth step, a management scheme was defined to formulate the scheduled biogas flows within an allowable fluctuation range[7]. On the plant side, a detailed operational strategy was implemented using an active adjusting mechanism, based on site-specific, data-driven analysis. On the user side, some timely incentives or regulatory measures could be used to influence individuals' willingness to use biogas. As for synergistic adjusting on both plant and user sides, we propose that additional flexibility in biogas flows, such as biogas storage optimization and professional operation, are also necessary to prevent unwanted biogas emissions, as biogas production and consumption will inevitably vary to some degree in practice[26].

### Biogas flow fitting and achieving carbon mitigation
Although the carbon mitigation potential of CBPD includes methane emission reduction of waste management and fossil fuel replacement, the contribution of manure management optimization was not calculated in this section, as the basic scenarios of manure treatment varied with different methane conversion factors, ranging from 0.1–80%[12]. Therefore, we first calculated the carbon mitigation of fossil fuel replacement to display the improvement of an upgraded CBPD. We presented five actual CBPDs in Chinese rural areas (see Supplementary Table 1 for detailed parameters) to investigate the status-quo of current operations, which supply biogas directly to rural inhabitants for household usage[7]. The biogas production capacities were designed according to the number of customers in each community, with a provision of 1 m³ biogas per customer per day. Their performance was used to provide insights into how to further realize the expected carbon mitigation potential of fossil fuel replacement. We investigated the regularity of biogas generation, the characteristics of biogas consumption, and carbon mitigation potential of fossil fuel substituting under different operational flexibility scenarios (Fig. 2).

The increase of biogas production rates had a lag phase of approximately 1.5 days after each feeding (Fig. 2a), while biogas consumption rate followed a daily repeating cycle (Fig. 2b), which was similar with that of general residential household's cooking[7], using biogas during normal mealtimes. In this case, current methane loss was assumed as 32.5%, which meant that carbon mitigation contribution of fossil fuel substituting had been eliminated. To turn things around, providing enough storage capacity and accurate CBPD operation are

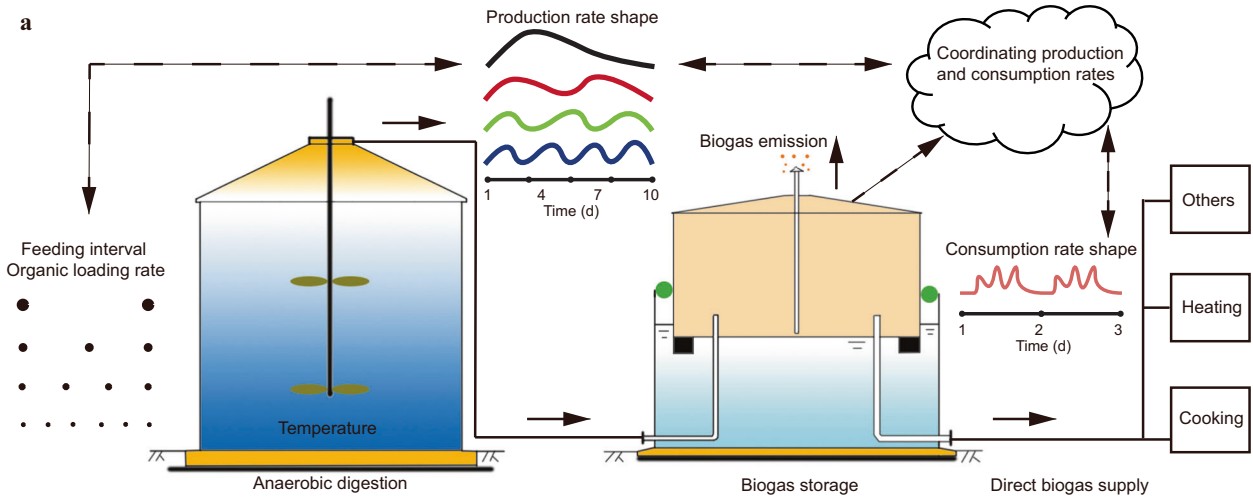

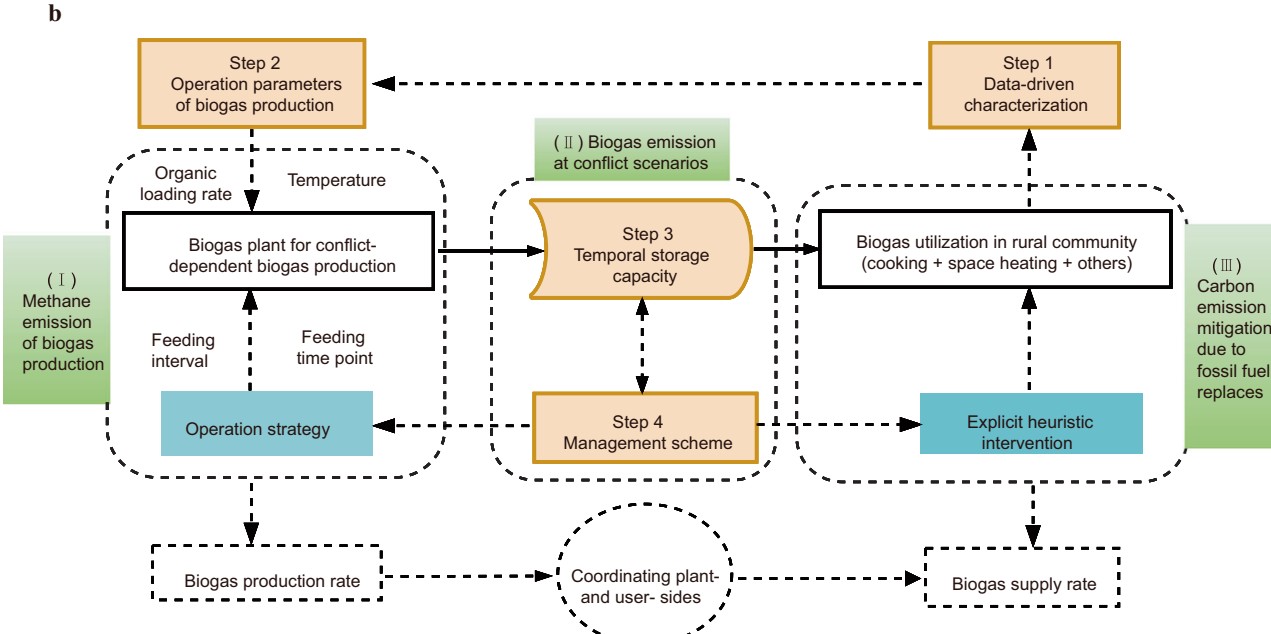

**Fig. 1 | Upgraded community biogas production and distribution system (CBPD) design. a** Schematic diagram of the biogas flow. Biogas losses occur when the temporal redundant biogas exceeds the storage capacity. **b** Framework to establish an upgraded CBPD for a demand-driven biogas supply. Data on the rural community's energy consumption is collected to estimate biogas consumption. The dynamically measured biogas supply rate is designed to equal the timely consumption/utilization rate in the community. Parts (I), (II) and (III) are sources of greenhouse gas emission or mitigation. Dotted arrows indicate information flow, and solid arrows indicate biogas flow.

feasible ways to achieve CPR = 1. On one hand, optimal storage capacity can efficiently resolve the conflict of production rate on the plant side and consumption rate on the user side. On the other hand, feeding with optimal amount and time points can synchronize the rates to reduce their mismatch. (see Supplementary Note 1). As Fig. 2c shows, the deployment of the proposed system would be converted current empirical operation with GHG emissions of 1.39 kg $CO_2$ eq $d^{-1}$ customer$^{-1}$ to a negative emission of −1.01 kg $CO_2$ eq $d^{-1}$ customer$^{-1}$. This indicated that only considering the contribution of fossil fuel replacement was not sufficient to realize carbon mitigation, and more attention should be directed to avoid biogas losses. Training operators with vocational skills to estimate the customer's biogas usage within accepting error, and establishing and implementing the optimal management strategies, are critically necessary to realize these goals. Taking the existing biogas consumption curve as an example, skillful manipulation can result in decreases of 11.6% and 9.5% for biogas

storage capacity requirements to achieve CPR = 1 at the scene of raising the consumption on the users side and decreasing the production on the plant side, respectively.

With sensitive analyses of the established data, 1.79 times biogas storage capacity amplification would realize the robustness and practicality of the proposed CBPD (Fig. 3a), which can avoid biogas losses in the most conflicted scenarios. Furthermore, it is highly recommended to establish an extra design of storage capacity in order to address other unexpected conflicts, defined as coordinated scenarios (Fig. 3b). For example, a fluctuation adjustment threshold can be added during the two continuous biogas production periods to better adjust the biogas supply design and avoid error accumulation (the error in one feeding cycle can be effectively eliminated in the next feeding cycle), because the situation where CPR is not equal to 1 in one feeding cycle is a possible state. This means the parameters of

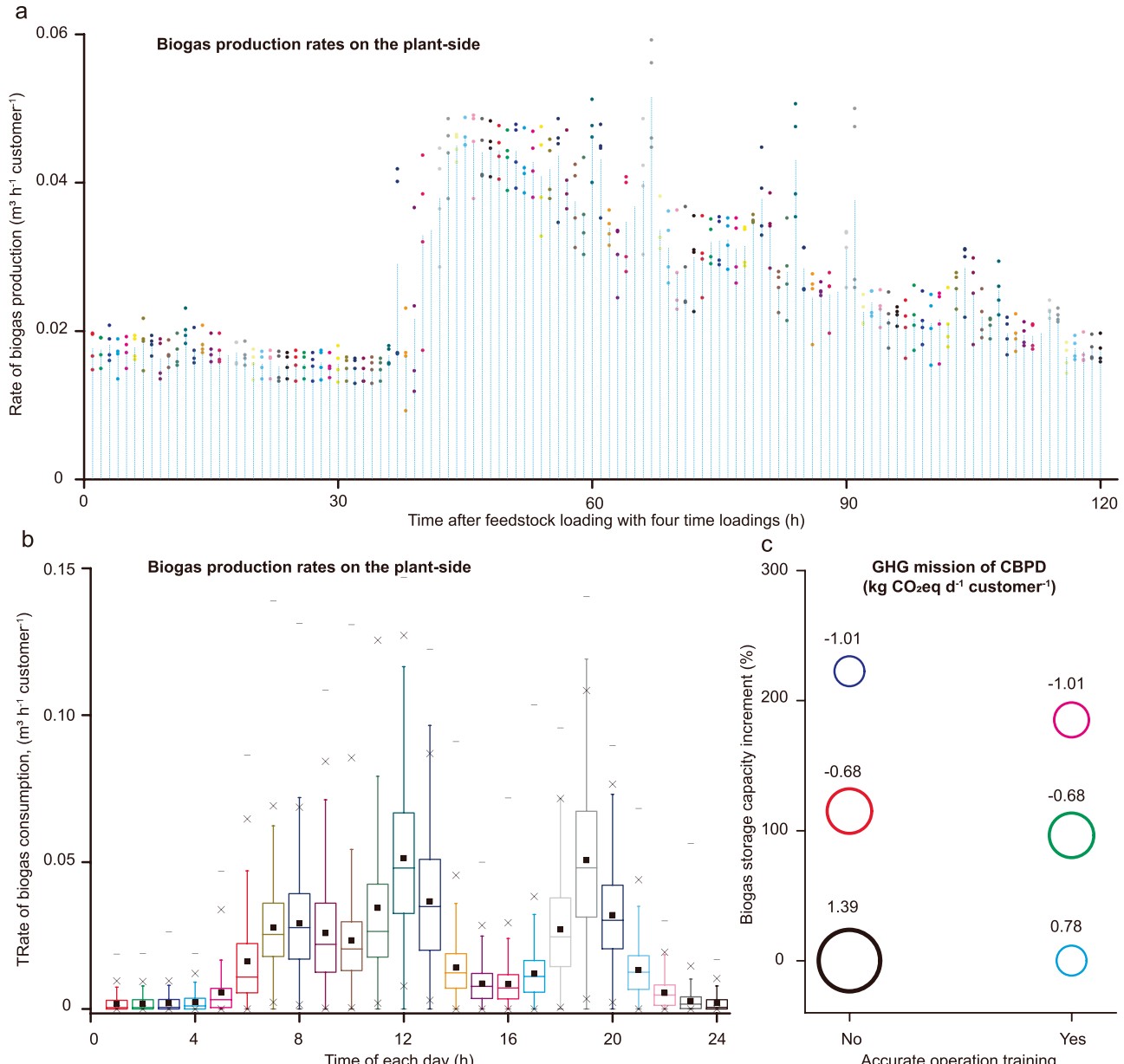

**Fig. 2 | Biogas production and consumption curve fitting and the greenhouse gas (GHG) emissions under different scenarios. a** Hourly biogas production with a feeding interval of 5 days. Continuous quality data (November 1–20, 2019) were selected to investigate the characteristics of biogas production rate curves. The points with one kind of color represent the data from corresponding feedings, and Arabic numerals show the average values at each time point. The biogas production rate was relatively steady during the initial stage (hours 0–38), but a sharp increase was observed during the middle stage (hours 39–48); after this peak, it declined gradually until the next feeding point (hours 49–120). **b** Customers' hourly biogas consumption in the five communities during the entire observation period (Aug. 8, 2017–Apr. 29, 2019). After the cleaning process, the number of qualified dataset is 991. The range of each box shows the interquartile range (IQR) of the distribution; the horizontal line inside the box shows the median value; and the Tukey whiskers extend to the farthest points of the distribution that are not outliers (i.e., those more than 1.5 × IQR from the edge of the box). The outliers are denoted by asterisks. **c** Six scenarios used to satisfy biogas on demand (colored bubble plots). The bubble size represents the energy-related and biogas leakage-related GHG emissions. The consumption-to-production ratio (CPR) of the current strategy is 67.5%, defined at the baseline condition, as shown in black. Operational training for feeding point optimization alone could provide more biogas on demand for customers' consumption, as shown in light blue. To achieve CPR = 1, two scenarios with substrate loading reduction on the plant-side are storage capacity optimization alone (shown in red) and a combination of storage capacity optimization and operational training (shown in green); and other two scenarios with increased customer consumption on the user-side are storage capacity optimization alone (shown in dark blue) and a combination of storage capacity optimization and operational training (shown in pink). The number at the top of each circle shows the value of GHG emission.

each feeding process should be adjusted (change in feeding point or amount) or maintained (a default operating process) for dynamic coordinating, which could be established on the basis of continuous information analyses of biogas consumption variations, residue biogas held in storage facilities, and biogas production consistency during the last feeding interval.

## State-of-the-art of upgraded CBPD during prevailing climate

Metrological and climatic conditions for various ambient temperatures and solar radiation intensities would play critical roles on the carbon mitigation of CBPD to some extent, and their combined influence could be calculated based on the outdoor solar-air temperature (Fig. 4a)[27]. Once it falls below the fermentation temperature, part of the

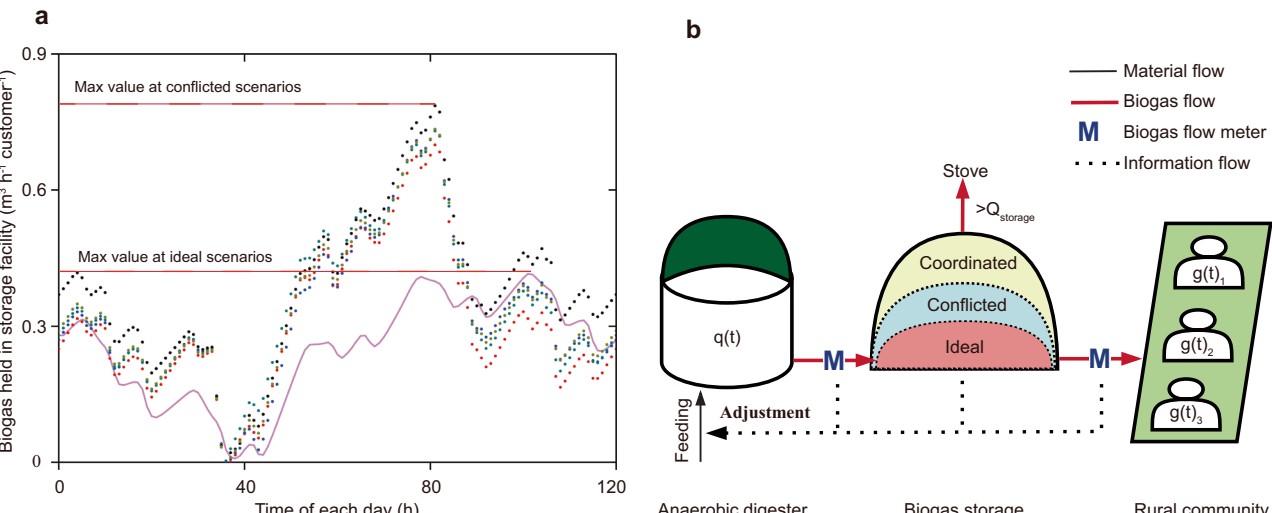

**Fig. 3 | Biogas storage design for synergistic adjustments. a** Variation of biogas held in storage facilities, with the given data during one feeding interval. The pink line is the ideal scenario, based on biogas production and consumption curve fitting; colored points show the conflicted scenarios of the selected five biogas production curves to fulfill biogas consumption under the most complex situations (the highest $R^2$ of the average biogas production of 5 continuous days). The total amount of produced biogas is assumed to equal that of biogas consumption during the feeding interval, and biogas held in the storage facility is the same at each feeding point. **b** Overview of biogas flow adjustment. Coordinated scenarios are assumed to add an allowable fluctuation range, considering the extra flexibility. Before each feeding, the operational information of the last feeding interval would be analyzed to modulate substrate amounts and feeding points. Once the gas held in the biogas storage facility reached the maximum capacity, extra biogas would be burned to heat the digester or other disposal facilities for emergency processing.

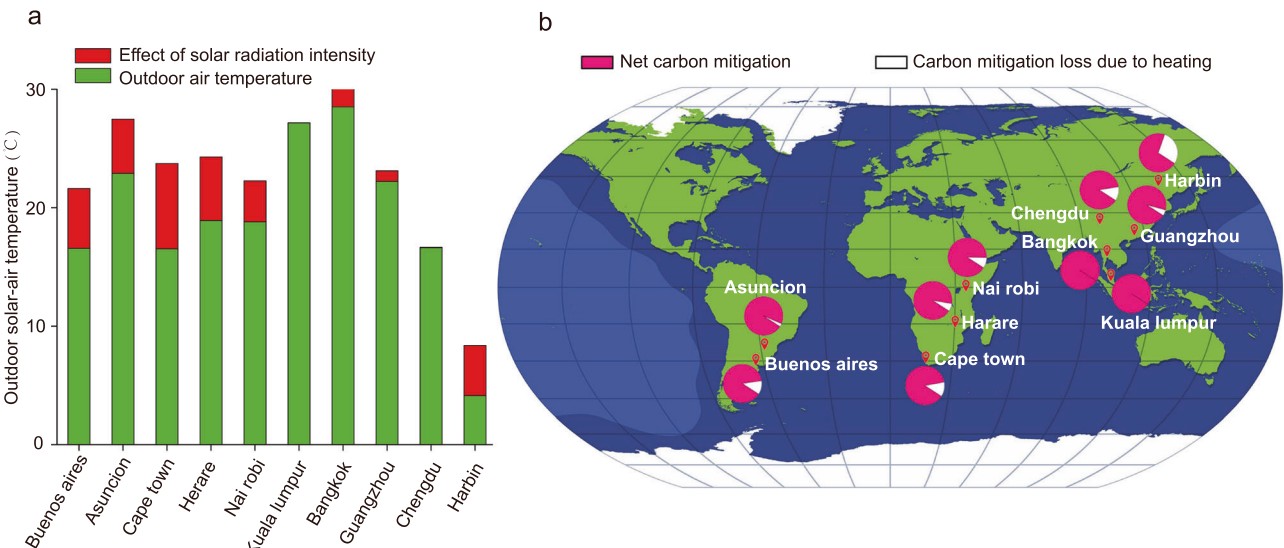

**Fig. 4 | Outdoor solar-air temperatures of 10 cities and net carbon mitigation of upgraded community biogas production and distribution system (CBPD) deployment. a** Outdoor solar-air temperatures. The annual averages of bulb temperatures and solar radiation intensities were used to calculate daily outdoor solar-air temperatures. **b** State-of-the-art decarbonization contributions of upgraded CBPD at different conditions. The amount of biogas supplied to customers was equal to the biogas produced minus the heating biogas used to maintain fermentation temperature. For representative calculations, Chinese rural communities' carbon mitigation of $-1.01\,kg\,CO_2eq\,d^{-1}\,customer^{-1}$ was used as the default value for biogas supply without heating requirements, and the values of net carbon mitigation were determined by the actual net biogas supply.

generated biogas should be used to compensate for the heat lost (the detailed calculation equation is provided in Supplementary Note 2). We used 10 cities in different developing areas to evaluate the performance of upgraded CBPD to reflect its universality; their performance of net carbon mitigation is shown in Fig. 4b (detailed parameters of the proposed CBPD calculation are shown in Supplementary Note 3 and Supplementary Table 2).

Upgraded CBPD deployment could acquire the highest decarbonization contribution to rural communities in the most iso-hyperthermic regions, such as Bangkok (Thailand) and Kuala Lumpur (Malaysia). Nevertheless, additional heating required for biogas production in Nairobi (Kenya) would decrease the carbon mitigation contribution by 9% owing to the low outdoor solar-air temperature. In frigid zones, low solar-air temperature result in low levels of net carbon emissions, such as, in Harbin (China), with $-0.38\,kg\,CO_2\,eq\,d^{-1}\,customer^{-1}$ during the day in January; nevertheless, it still has a greater carbon mitigation contribution than that of the widely applied combined heat and power generation unit (CHP) in developed areas (Supplementary Table 3)[28]. As for the other six cities, GHG emissions were $-0.88$ to $-0.95\,kg\,CO_2\,eq\,d^{-1}\,customer^{-1}$. These indicated that use

of upgraded CBPD had a significant advantage on carbon mitigation achievement. Furthermore, it could have a greater feasibility than that of CHP, which normally has a highly volatile operation revenues, depending on the real-time electricity prices for spatiotemporal variations and the general longer transport distance[29]. Considering the above, upgraded CBPDs have the potential to be widely used as scalable decarbonization solutions in developing economics. Taking China as an example, national deployment of upgraded CBPD in China could have the potential to eliminate 62.4% carbon emissions of rural inhabitants[30].

**National deployment to meet the Chinese 1.5 °C warming target**
Although diverting organic waste through anaerobic digestion is the most effective approach toward net-zero warming among the mainstream technologies, the feasible systems in lower middle-income and low-income countries should be simple and practical, which would mainly include a family size household digester, CBPD, upgraded CBPD, and so on[31]. After comparison and analysis (see Supplementary Table 4), upgraded CBPD could have the highest potential to efficiently utilize biogas and mitigate carbon for board application in current Chinese rural area. Because of the intensive management of animal breeding and rural inhabitants living in scattered communities[32], it could provide the viable scenario for feedstock collection on the plant side and a certain amount of biogas requirement on the user side due to the relatively fixed customer group for upgraded CBPDs' deployment (for detailed instructions on implementation, see Supplementary Note 4).

As animal waste is an excellent, cheap, and highly accessible feedstock material[33], its local availability is the most essential factor for broad use. Thus, availability of the most common manures (from pigs, chickens, and cattle) at the provincial level was investigated to evaluate the feasibility for national deployment (Fig. 5a). Generally, Chinese mainland areas are likely candidates for the broad application, as the rates of methane production potential from manure and domestic biogas demand in the rural regions (RPD) are high in most provinces. Shanghai is the only province, whose RPD is less than 1, and co-fermentation with another organic waste is therefore the potential path to fulfill biogas consumption[15]. Meanwhile, Shanghai is a relatively small-population province with a low proportion of rural population, accounting for 0.7%, and its impact on the achievable targets of national deployment will probably be negligible. Even so, quantifying the characteristics and recovery percentages of different manure sources are still important tasks to determine the feedstock collection strategy, due to the large spatial and temporal heterogeneity in the associated biogas production potential[12]. Furthermore, it also should be noted that long distances of feedstock transportation could decrease the carbon mitigation contribution, owing to a large amount of fossil fuel use[34]. In addition, feedstock logistics would also be influenced by stakeholders' revenues, transportation costs, and public interest at different rural communities[29], as the feedstock supply chain is a market interpretation. Thus, it is highly essential to identify the region- and practice-specific feedstock collecting strategy before the on-site CBPD is deployed, including exploring specific models of biogas production rate of segregated feedstock for co-fermentation with other kinds of agriculture waste[15], and properly managing feedstock logistics to attain the assumed biogas production. In addition, it is also important to clarify the coordinated scenarios of biogas storage capacity design for different feedstock's feeding[35].

Apart from the contribution of energy replacement, we were also interested in its effect (with CPR = 1) on methane emission reduction in manure management, which is one of the main methane emission sources in rural areas[13]. As there were still 186.7 million rural households in China in 2020[36], most of which had no access to natural gas[37], a national upgraded deployment of CBPD could provide an alternative path to achieve energy equality and may contribute 3.77% to meet the Chinese carbon mitigation commitment of the Paris Agreement of a 1.5 °C increase in global warming (Fig. 5b), containing both methane emission reduction of manure management and carbon mitigation of fossil fuel substitution[38], which was calculated by using the recent Chinese report submitted to the United Nations[30].

As Fig. 5c shows, current rural inhabitants' biogas usages only account for 25.4% of manure's methane production potential at the investigated biogas consumption rate, according to the calculation given above. An industrial or scalable level of biogas use is essential to supply more clean energy to the rural communities, by exploring diverse paths of biogas consumption and adjusting the specific biogas flows. Although the direct supply power to communities has the lowest environmental performance due to the relatively low energy efficiency in converting biogas to electricity (see Supplementary Table 3), it still has a remarkably positive effect on climate change. Even if considering the contribution of fossil fuel substitution only, further biogas usage from manure conversion still can have a vital extra GHG abatement of national fossil fuel combustion, ranging from 0.9–1.7%[30].

## Discussion
We proposed a simple and practical strategy to modulate biogas flow of CBPD for CPR = 1, which can maximize energy and climate benefits in rural developing areas, rather than injecting biogas or biogas-based natural gas into centralized grids[15]. The upgraded CBPD could be formed by acquiring the kinetic parameters associated with the plant and user sides, designing a flexible interaction, and establishing coordination-adjusting mechanisms for each feeding process. We also showed that upgraded CBPD could produce sustainable energy at prevailing climate conditions, and proposed that its broad application could have a vital positive effect on climate change mitigation, organic waste management optimization, and fossil fuel offsetting, while avoiding the credits for energy supply on demand and biogas losses. Regarding the energy consumption of Chinese rural inhabitants, cooking is the largest use, accounting for 41.6% of the total[39], so upgraded CBPD could facilitate a transition towards ever wider use of biogas. In addition, energy consumption in rural communities will certainly increase until 2030, due to Chinese demographic changes[2,40], and the deployment of upgraded CBPD could further increase biogas use in the fields of residential living and agriculture production. It also could provide a substantial incentive to pursue a path toward the reduction of fossil fuels, and reverse the current trend of carbon emissions increases in the Chinese countryside[2]. Such policy options must be evaluated in more detail to convince investors that upgraded CBPD is a reliable technology, and increase targeted subsidies for a combination of storage capacity optimization and operational training.

Feedstock availability and their characteristics are variable, owing to spatiotemporal differences. Advanced algorithms and data analysis techniques are efficient tools to create more positive impacts on quantifying biogas production variance (time requirement) and optimizing operation strategies to form beneficial products in realistic cases. A specific feedstock collecting plan should be identified before the upgraded CBPD deployed, as biogas supply on demand is significantly correlated with time dependency and transportation limits of organic waste management[15]. A limitation of this study was that RPD analysis was conducted at the provincial scale. Once it is investigated at small regional levels, such as towns, a more accurate targeted result could be obtained. To attain maximum energy and climate benefits in practical, we highly recommend practical innovations on regional animal husbandry and manure collection for high-quality performance of biogas generation[41], as the current amounts of poultry and livestock farming are increasing and manure management strategies are varied in China[42].

Upgrading the deployment of the CBPD would be carried out in rural communities in a decentralized manner. This could be conducted to store and supply organic fertilizer on-site for distribution distances minimization, to decrease direct nitrous oxide emissions by avoiding

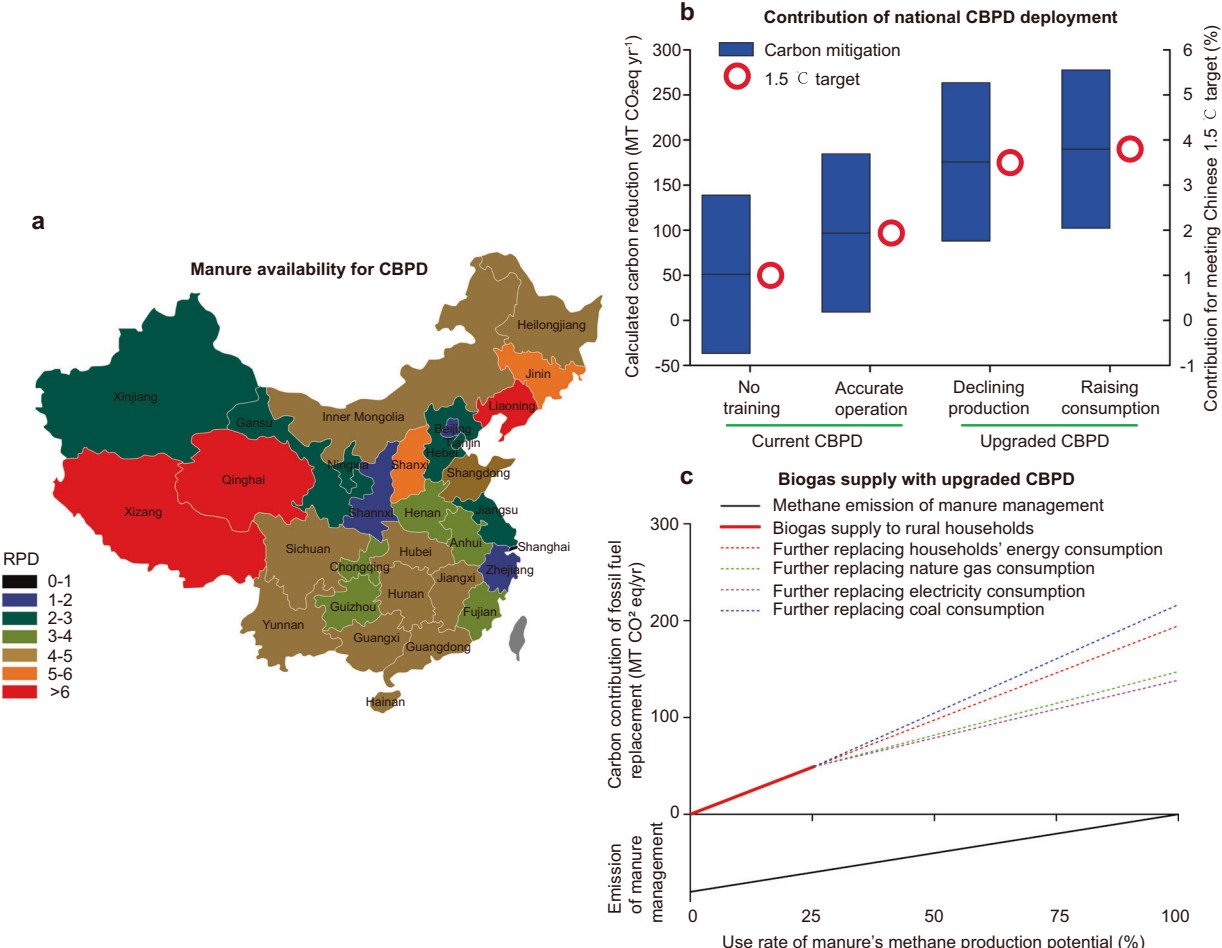

**Fig. 5 | Manure availability for national deployment of upgraded community biogas production and distribution systems (CBPD) in rural China and its potential contribution. a** The availability of manures (pigs, chickens, and cattle) in each province (omitting Taiwan, Macao, and Hong Kong). For the detailed rates of methane production potential from manure and domestic biogas demand in the rural regions (RPD) at the provincial level, see Supplementary Table 5. **b** Carbon emission reductions of national CBPD deployment. It is equal to the assumed emission of CBPD deployment minus that in baseline scenarios. The baseline scenarios are manure treatments without a biogas collection and utilization system, and the pit storage with lower limits (methane conversion factor value of 0.2) and reservoirs as anaerobic deep lagoons (methane conversion factor value of 0.8) were selected to represent the best scenarios and the worst

scenarios practical, respectively[12]. The assumed scenarios are national CBPD deployment at the investigated biogas consumption level of rural inhabitants, number of which is 1.10 million households, excluding township population. Chengdu City's climate parameters were taken as the overall average value in China, with 11.3% of the produced biogas being used to maintain the fermentation temperature. The black horizontal lines represent the average carbon reduction potential with respect to the assumed baseline scenarios with the methane conversion factor of 0.5, and the blue bar area means the fluctuating range under the possible scenarios. **c** Exploring manure's full methane potential with upgraded CBPDs under different usage scenarios. The dotted lines are colored based on the further pathways of biogas utilization, except for the national deployment for the existed domestic biogas demand.

denitrification processes[12], and reduce synthetic fertilizer inputs, because more manure nitrogen is recycled back to farmland instead of being released into the open air[43]. These could facilitate ecological utilization of an anaerobically digested fertilizer to nearby farms[44]. This could also mean that once all processes can be regionally integrated well, such as biogas consumption, breeding intensity and form, and planting structures, the benefits of circular agriculture realization may be achieved with the intrinsic synergistic interaction[45]. Thus, rural communities in developing areas are encouraged to achieve energy equality and regional carbon mitigation by deploying upgraded CBPDs, because of the high feasibility of biogas demand and supply matching and efficient nutrient circularity of manure[46].

The methane production potential of manure in China is 2467.7 billion MJ yr$^{-1}$, which only covers 26.3% of current direct residential energy consumption in rural areas[47]. In addition, rural manufacturing industries would provide an exponential of energy consumption due to China's rural revitalization strategy[48]. The huge energy demand and its clean energy renovation of solid fuel substituting also requires

more upgraded CBPD deployment for converting various agricultural wastes to satisfy time-use demand in rural areas. Except for upgraded CBPD design and skillful manipulation, a potential strategy leading to a more resilient system is to dynamically adjust biogas usage by kinetic parameters learned from detailed applications. Further research should focus on auxiliary measures to minimize the requirement of biogas storage volume and the negative impact of undesirable processes (for example, temporal congestion caused by inaccurate feeding time points or amount selections without intervention), which contain the policy of time-of-use pricing at peak and trough biogas usage[29]. In this way, energy usage in future scenarios can be accurately predicted, and biogas supply can be planned for another usages, such as, manufacturing industries.

Upgraded CBPD illustrates a way to stable supply biogas in dynamic situations with full utilization, and its broad feasibility could acquire an important contribution of climate change mitigation. Further policy measures should include designing administrative regulation to cultivate professional units to provide reliable biogas supply

service, and effective judgment criteria for methane mitigation subsidies based on institutional designs using Common Pool Resource theory[49]. In addition, upgraded CBPD deployment in combination with other renewable energy systems, such as solar energy, heat pumps, etc., could develop superior synergistic systems for more stand-alone energy supply. It could acquire a higher primary energy use ratio by more efficiently applying the characteristics of upgraded CBPD's flexible energy production[50].

## Methods

### Biogas flow and biogas storage capacity

If the amount of biogas held in a storage facility is $Q(t)$, then it is $Q(t+\triangle t)$ at time $t+\triangle t$, where $\triangle t$ is time interval. On the basis of mass balance, $Q(t+\triangle t)$ should theoretically be equal to the sum of $Q(t)$ and net variation in biogas amount during the time interval $\triangle t$, as shown in Eq. 1.

$$
\begin{aligned}
Q(t+\Delta t) = \int_t^{t+\Delta t} q(t)dt - \int_t^{t+\Delta t} g(t)dt \\
- \int_t^{\Delta t} e(t)dt - \int_t^{\Delta t} h(t)dt - Q(t)
\end{aligned}
\tag{1}
$$

Where: $q(t)$, $g(t)$, $e(t)$, and $h(t)$ are the biogas production rate, biogas consumption rate, biogas emission rate, and the biogas required rate to generate heat for fermentation temperature maintaining in cold weather, respectively.

The biogas consumption rate is the accumulated value of all customers' usage, as shown in Eq. 2.

$$
\int_t^{t+\Delta t} g(t) = \sum_i g(t)_i \cdot N_{customer,i}
\tag{2}
$$

Where: $g(t)_i$ is average biogas consumption rate of customer type $i$ in the region; $N_{customer,i}$ is the number of customer type $i$ in the region.

The biogas storage capacity design accounts for the sum of maximal storage capacity and requirement of margin design, as shown in Eq. 3.

$$
Q_{storage} = k \times Max(Q(t) : Q(t+t_{feeding\ interval}))
\tag{3}
$$

Where: $Q_{storage}$ is biogas storage capacity; $k$ is safety factor for margin design to avoid biogas emission; $Max(Q(t):Q(t+t_{feeding\ interval}))$ is the maximal value of residual biogas held in biogas storage facility during the assumed period.

### Data analysis of CBPD

Hourly biogas production, hourly biogas consumption, and their methane contents of the five selected CBPDs were measured using transit-time ultrasonic gas flow meters (TY1030, TianYu, Wuhan, China), which were easy to install with minimal or no disruptions to the flow, and had several vital advantages, such as high accuracy and a wide range ratio[51]. Each CBPD was equipped with two meters with dehydration systems, which were used to avoid dew formation[52]. One was installed before the biogas storage facility to measure the production flow, and the other was on the main pipeline of biogas supply. All meters were calibrated and validated at test facilities for biogas measurement every 6 months.

All data, $(X_1, X_2,..., X_n)$, were restructured to achieve schema integration of the feeding interval data, including steps such as splitting, merging, folding, and unfolding, to resolve and overlap conflicting representations. The measured biogas production set and biogas consumption set were represented by sets $S_p$ and $S_c$, respectively; $S = \{X_1, X_2,..., X_n\}$. The daily biogas production set and biogas consumption set were represented by sets $X_p$ and $X_c$, respectively; $S = X(n, t) = \{x_1, x_2,..., x_{24}\}$. The data collected were used as the respective X at day n and hour t.

To detect and remove the sets of errors and inconsistencies, a detailed data analysis was performed. The cleaning process on the given dataset made the following assumptions.

1. If any value of $X_p$ was less than 0.1 or more than five times the daily average value, the value was either considered to be an outlier or biogas production did not follow the normal distribution, and day $X_p$ was removed from the dataset.

2. If any value of $X_c$ between 1 am–4 am was more than 0.5 m³ h⁻¹ customer⁻¹, it meant that biogas leakage or an inaccurate measurement may have occurred. Furthermore, if the daily average value of any $X_c$ was two times higher or 0.5 times lower than that of the previous or following day, the data for the daily biogas consumption rate was considered to be an outlier, and the day $X_c$ was removed from the dataset.

3. Finally, only when day $X_p$ and $X_c$ were both in the dataset, could the values be considered to be quality data; otherwise, the single X value was deleted.

### GHG emission calculations for the CBPD

In this study, GHG emissions of CBPD at a time interval of $\triangle t$ ($E_{CBPD}$, kg $CO_2$eq d⁻¹ customer⁻¹) accounted for utilizable biogas and methane slipping due to biogas emissions into the open air, as shown in Eq. 4.

$$
\begin{aligned}
E_{CBPD} = 25 \cdot \Big( \big( X(\Delta t)_p - X(t)_p \big) - \big( X(\Delta t)_c - X(t)_c \big) \Big) \cdot r \cdot \rho \\
- \sum_j (X(\Delta t)_c - X(t)_c) \cdot r \cdot C_j \cdot E_j
\end{aligned}
\tag{4}
$$

Where: 25 is the conversion factor of a methane emission to $CO_2$ equivalents; $r$ is the methane content of biogas (%); and $\rho$ is the density of biogas, which is taken at 20 °C and 1 atmosphere pressure and has a value of 0.67 kg m⁻³; $C_j$ is the fraction of biogas used to substitute energy type $j$; $E_j$ is the collective $CO_2$ emission factor of energy type $j$ use (see Supplementary Table 6), the assumed carbon emission value of rural households' energy use in this study was 0.0739 kg $CO_2$ eq MJ⁻¹, which is equal to the collective emissions of the rural residential energy weighted mean value, calculated by the energy structure and corresponding emission factors[53,54].

Net GHG emissions of upgraded CBPD on site at time interval of $\triangle t$ ($E_{CBPD,\ net}$, kg $CO_2$eq d⁻¹ customer⁻¹) accounted for net biogas production, as shown in Eq. 5.

$$
E_{CBPD,net} = - \Big( \big( X(\Delta t)_p - X(t)_p \big) - \frac{Q_T}{21.54 \times 0.7} \Big) \cdot r \cdot C_j \cdot E_j
\tag{5}
$$

Where: $Q_T$ is biogas required for fermentation temperature maintaining at time interval of $\triangle t$ (MJ); 21.54 is the caloric value of biogas containing 60% methane (MJ m⁻³); 0.7 is the heat efficiency of biogas conversion and heat exchange for maintaining fermentation temperature[55]. The thermal balance calculation of $Q_T$ is described in Supplementary Note 2.

### RPD

RPD is equal to the methane production potential of available manure divided by the total rural biogas demand in the certain area, as shown in Eq. 6.

$$
RPD = \frac{\sum_1^n N_n \cdot VS_n \cdot B_{0,n}}{N_{household} \cdot S_{average} \cdot 0.6}
\tag{6}
$$

Where: $N_n$ is the daily collected manure from a livestock species/category n in the region (tone d⁻¹); $VS_n$ is the volatile solid of livestock's manure in category n (tone tone⁻¹); $B_O,n$ is the maximum methane production potential of manure produced by livestock in category n (m³ $CH_4$ tone⁻¹); $N_{household}$ is the number of rural inhabitants in the region; $S_{average}$ is the average daily biogas consumption of the rural inhabitants (m³

biogas d$^{-1}$); and 0.6 is the recognized methane content of biogas (m$^3$ CH$_4$ m$^{-3}$ biogas). $N_n$ was obtained from China Agriculture Yearbook[6], $VS_n$ and $B_O,n$ were obtained from the 2019 Refinement to the 2006 IPCC Guidelines for National Greenhouse Gas Inventories[12], $N_{household}$ was obtained from a national population census[36], and $S_{average}$ was calculated from the data of the five CBPDs examined in this study.

**Methane mitigation calculations of manure management optimization**

The amount of methane emission from a manure management system in the region (CH$_4$ emission) is expressed as in Eq. 7.

$$CH_4\, emission = \sum_1^n N_n \cdot VS_n \cdot B_{0,n} \cdot MCF \qquad (7)$$

Where: $MCF$ is methane conversion factor of specific manure management system, representing the degree to which $B_O,n$ is achieved. In this study, the default value of methane emission of upgraded CBPD is assumed to be 0.

Calculated carbon mitigation relative to the baseline scenario (CMB) of national deployment and its contribution for meeting the Chinese 1.5 °C target ($RT_{1.5°C}$) is expressed as in Eqs. 8 and 9.

$$CMB = CH_4\, emission - N_{household} \cdot S_{average} \cdot 21.54 \cdot 0.0739 \qquad (8)$$

$$RT_{1.5℃} = \frac{CMB}{0.45 \cdot T_{emission}} \qquad (9)$$

Where: 0.45 is the requirement of carbon emission decrement to meet the 1.5 °C global warming target according to the sixth assessment report of IPCC[38]; $T_{emission}$ is the total carbon emission in the region.

Carbon contribution of fossil fuel replacement ($E_{replace}$) is expressed as in Eq. 10.

$$E_{replace} = \sum_{i,j} g(t)_i \cdot N_{customer,i} \cdot C_j \cdot E_j \qquad (10)$$

**Reporting summary**

Further information on research design is available in the Nature Portfolio Reporting Summary linked to this article.

## Data availability

Main data supporting the findings of this study are available within the Manuscript and Supplementary Information. Source data underlying figures in the Manuscript are provided within the Source Data. Source data are available at the figshare repository.

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

## Acknowledgements

T.L. gratefully acknowledges financial support from the Sichuan Science and Technology Program (Nos. 23ZDYF0252 and 2023YFS0386), the Central Public-interest Scientific Institution Basal Research Fund (No. 1610012022008_03102), and the Agricultural Science and Technology Innovation Project of Chinese Academy of Agricultural Sciences (CAAS-ASTIP-2021-BIOMA).

## Author contributions

T.L. conceived the paper, wrote the code, and drafted the manuscript. B.S. conceived the framework and made improvements to the manuscript. Z.M. conducted the case data recording and data analyses. A.H. made improvements to the manuscript. K.J. wrote the relevant part of the Methods. All authors approved the final manuscript.

## Competing interests

The authors declare no competing interests.
