## [Peer Review File · Nature Communications]

Unlocking the full potential of biogas systems for sustainable energy production and climate solutions in rural communitiesReviewers' Comments:

Reviewer #1:

Remarks to the Author:

The authors have developed an upgraded community biogas production and distribution system (CBPD) to achieve a nearly 1:1 ratio of biogas consumption to production in rural or remote areas, effectively addressing the issue of energy equality. Their upgraded CBPDs have the potential to reduce carbon emissions in China by 3.77%, aligning with the 1.5°C global warming target of the Paris Agreement. This aspect of the article is highly intriguing and holds the potential to garner broad interest.

However, upon in-depth examination of the article, several significant questions and concerns emerge, which should be addressed carefully.

Introduction: There are multiple mentions of the importance of reducing methane emissions. While it's a crucial point, it could be mentioned once and then referred to when discussing the benefits of the CBPD system.

In the introduction, data about methane production potential is mentioned, but it's not entirely clear how this data was obtained or its reliability. It would be helpful to provide a brief explanation or source.

The text in the introduction could benefit from a clearer, more focused problem statement at the beginning. It should explicitly state that the primary issue is the need for an efficient and widespread biogas distribution system that mitigates methane emissions.

CBPD optimization for demand-dependent biogas production and distribution: If any specific models or equations are used in this section, it may be helpful to reference them for readers who want to explore them further.

National Deployment to Meet the Chinese 1.5°C Warming Target: This section highlights the potential of decentralized upgraded CBPD to utilize animal waste as a readily available feedstock material. It effectively addresses the importance of local availability and its impact on broad CBPD deployment. However, the text could benefit from a more in-depth exploration of two vital aspects. Firstly, it would be valuable to delve into the challenges and solutions related to slurry management, a critical factor that can influence rural communities' willingness to adopt biogas systems. Effective strategies for efficient slurry management are key to mitigating one of the primary challenges faced by potential users.

Secondly, the text mentions that most of the Chinese mainland areas, with their high rates of methane production from manure and high domestic gas demand in the rural community (RPD), are likely candidates for the broad deployment of upgraded CBPD. Did the authors consider the changing landscape of rural areas, with livestock production moving closer to cities due to transportation cost considerations? This raises important questions about the adaptability and reach of biogas systems in evolving rural communities, particularly those that may no longer be considered purely rural areas. Further discussion on these aspects in the "Discussion and Policy Implications" section would provide a more comprehensive understanding of the biogas landscape's dynamics in rural China.

Discussion and policy implications: While this discussion outlines the promising pathway for upgrading CBPD systems and achieving energy and climate benefits, it's important to acknowledge the challenges related to biogas emissions and the technical expertise required for successful implementation. Addressing the availability of skilled technicians and tackling biogas emission complexities in the context of rural communities should be a critical consideration in further research and policy development. These challenges are pivotal in ensuring the practical feasibility and success

of the proposed strategies.

Reviewer #2:

Remarks to the Author:

The authors explored the potential of an upgraded Community Biogas Production and Distribution system (CBPD) to meet households' cooking energy needs and mitigate greenhouse gas (GHG) emissions as an alternative to natural gas. Anaerobic digestion (AD) technology holds significant promise for providing clean energy for cooking, heating, and lighting, while simultaneously addressing indoor air pollution and organic waste management issues in rural areas. To fully realize its potential, widespread adoption of AD technology in rural settings is indispensable. The extensive uptake of AD technology as an energy producer hinges on ensuring both the reliable supply of biogas and the environmental and economic benefits of biogas as an energy source. In this context, the present study underscores the potential of the upgraded CBPD to ensure a demand-driven biogas supply while minimizing GHG emissions resulting from biogas slippage.

General comments:

The methodology section could benefit from greater detail. The transparency of the developed model and data analysis could be enhanced if the authors shared the collected data and the specifics of the developed model.

The discussion presented in this study is solely derived from the current investigation. The authors did not undertake a comparative analysis or discussion of results in the context of related studies. While a direct comparison is not feasible due to the absence of a specific study, the authors could have drawn insights from different research endeavors exploring temporal and spatial variations in feedstock availability for biogas production, the potential for biogas production, current and projected household energy demands, and the greenhouse gas mitigation potential of biogas.

The study calculates biogas production potential based on the total livestock number and their manure production potential. However, it is crucial to note that not all animal manure is necessarily available for biogas production. The recovery percentage of manure varies depending on factors such as animal types, location, and livestock production systems. It remains unclear whether the authors have factored in the recovery percentage of animal manure in their calculations of biogas production potential. Clarification on this aspect would strengthen the validity and applicability of the findings.

Crop residues have also been considered as a potential feedstock for biogas production. Crop residues are indeed a significant potential feedstock for biogas production, and it's crucial to consider their major types, availability, and biogas production potential. The data collection and analysis section of the paper, however, does not address these aspects. Moreover, crop residues not only differ in their biogas production potentials but also in their lag phase and production rates. A more detailed analysis would have significantly benefited the paper.

The authors have focused on calculating the greenhouse gas emissions mitigation potential of biogas as a substitute for natural gas in meeting rural households' energy demand for cooking. However, the paper acknowledges that natural gas is not readily available in all rural areas. Rural areas without easy access to clean energy options commonly rely on conventional sources such as firewood, crop residues, and dried animal dung for their cooking and heating needs. Therefore, calculating greenhouse gas emissions mitigation potential based solely on natural gas substitutions may not fully reflect the actual energy practices in rural areas.

The calculations in the study are based on the current household energy demand for cooking and the biogas production potential of animal manure. However, it is acknowledged that both the energy

demand for cooking and the amount of animal manure available for biogas production are subject to change over time. It would be valuable for the developed model to incorporate sensitivity analysis to project the greenhouse gas mitigation potential of the CBPD under different scenarios, accounting for potential variations in energy demand and feedstock availability over time.

Furthermore, the study maintains a constant digester volume of 100 m³ throughout the calculations. The rationale behind selecting this specific volume is not clearly explained. It's essential to clarify how the chosen digester volume aligns with the available feedstock (i.e., animal manure and crop residues) and total energy demand for cooking, which can vary across different rural settings. Providing a clear explanation for selecting a particular digester volume and illustrating how it accommodates variations in both feedstock supply and energy demand would enhance the manuscript.

Including sensitivity analysis in the study would be beneficial. For instance, assessing the sensitivity of the estimated greenhouse gas mitigation potential to factors such as the size of the digester, storage tank, or biogas demand would provide valuable insights. If the results are found to be less sensitive to the size of the biogas storage tank, considering a slightly larger tank may better ensure a stable biogas supply in the event of unintended sudden fluctuations in biogas production or demand. This consideration would contribute to the robustness and practicality of the proposed CBPD system.

Specific comments:

The manuscript requires thorough revision for both English language usage and technical terminologies.

Line 128: Should it be "1 m³ of biogas per customer per day"?

Lines 145 & 146: Please provide the unit for GHG emission values.

Line 193: Please define RPD.

Lines 210-241: Further elaboration and clarification are needed. In its current form, it is unclear whether the authors are presenting their own findings or stating literature values.

Lines 231-236: In rural areas, animal manure is typically applied to agricultural land as Farmyard Manure (FYM). This implies that even in the absence of the proposed CBPD, animal manures will find their ultimate fate in agricultural land. Therefore, the authors may need to explain the added merits of digestate over FYM rather than simply overemphasizing the environmental merits of digestate.

Line 294: Please check for "RPD" and "PRD."

In Supplementary Table 1, the authors have mentioned CPR values of less than 80% for all the rural areas studied. Since households are using natural gas and firewood to meet their cooking energy demand, are such CPR values less than 1 due to a disparity in the production and consumption of biogas or due to the higher cost of biogas compared to firewood or the convenience of using natural gas?

In Supplementary Table 2, please specify whether the OLR value is based on a wet weight or dry weight basis.

In Supplementary Figure 1, please explain the meaning of "biogas residue."

Reviewer #3:

Remarks to the Author:

The manuscript Titled ""Unlocking the full potential of biogas systems for sustainable energy production and climate solutions in rural communities" presents a valuable contribution to the field of sustainable energy, particularly in the context of rural biogas systems. The proposed CBPD system has the potential to make a significant impact in terms of energy equality and carbon emissions reduction. However, the study would benefit from a more in-depth analysis of the methodologies, a critical evaluation of its limitations, and a more direct connection between the findings and policy implications.

Abstract and Introduction

– The abstract concisely outlines the objective and scope of the study, highlighting the development of an upgraded community biogas production and distribution system (CBPD) and its potential impact on carbon emissions reduction. However, it could benefit from a more detailed explanation of the methodologies and key findings

– The introduction effectively sets the context for the study, emphasizing the significance of biogas as a sustainable energy source. However, a more comprehensive discussion of the current challenges in biogas system implementation, particularly in rural contexts, would strengthen the background
Methodology

– The methodology section provides a clear description of the steps involved in optimizing the CBPD system. It includes the identification of biogas demand rate, quantification of operational parameters, design of biogas storage capacity, and determination of operational strategy.

– The data analysis process is well-detailed, especially in handling data inconsistencies and errors. However, the manuscript could benefit from a more in-depth discussion on the choice of methods and tools used for data collection and analysis, particularly the use of multipath ultrasonic gas flow meters

Results and Discussion

– The results section presents a comprehensive analysis of the biogas production and consumption patterns. However, there is a lack of detailed statistical analysis or comparative study to validate the effectiveness of the proposed CBPD optimization strategies.

– The discussion effectively highlights the potential impact of CBPD systems on energy equality and carbon emissions reduction in rural communities. However, it could be enhanced by including comparisons with existing biogas systems and discussing the scalability and practical challenges of implementing the proposed system.

Conclusions and Policy Implications

– The manuscript concludes with significant findings, emphasizing the potential of CBPD systems in reducing carbon emissions and achieving energy equality in rural areas. However, the conclusions could be more impactful by summarizing key findings more explicitly and suggesting future research directions.

– The policy implications section is insightful, focusing on the need for improved managerial skills and the integration of CBPD with other renewable energy systems. A more explicit connection between the research findings and specific policy recommendations would be beneficial.

General Comments

– The manuscript is well-structured and presents a novel approach to optimizing CBPD systems. However, it would benefit from a more critical analysis of the limitations of the study and potential barriers to implementation.

– The figures and tables provided are relevant and support the text well, but they could be more effectively integrated into the narrative to enhance the reader's understanding.

– The references are comprehensive, but there is room for including more recent studies to strengthen the manuscript's grounding in the current state of research in this field

Reviewers' comments:

Reviewer #1: (marked in red for color highlighting in the revised manuscript)

– Comment 1:

The authors have developed an upgraded community biogas production and distribution system (CBPD) to achieve a nearly 1:1 ratio of biogas consumption to production in rural or remote areas, effectively addressing the issue of energy equality. Their upgraded CBPDs have the potential to reduce carbon emissions in China by 3.77%, aligning with the 1.5°C global warming target of the Paris Agreement. This aspect of the article is highly intriguing and holds the potential to garner broad interest. However, upon in-depth examination of the article, several significant questions and concerns emerge, which should be addressed carefully.

Response: (marked in red in the revised manuscript)

- Thanks very much for your positive comment. We have addressed the listed questions and paid more attention to the relative concerns accordingly, the details will be found in the following. Please check.

– Comment 2:

Introduction: There are multiple mentions of the importance of reducing methane emissions. While it's a crucial point, it could be mentioned once and then referred to when discussing the benefits of the CBPD system.

Response: (marked in red in the revised manuscript)

- Thanks very much for your valuable suggestion. We have discussed the methane mitigation contribution of the CBPD system accordingly. Firstly, we have pointed out that the carbon mitigation potential of CBPD includes methane emission reduction of waste management and fossil fuel replacement; sequentially, we calculated the carbon mitigation of fossil fuel replacement to clearly illustrate the improvement of an upgraded CBPD at the first step, as the basic scenarios varied with the methane conversion factors ranging from 0.1–80%, which are added in the section “Biogas flow fitting and achieving carbon mitigation”. Secondly, we discuss the methane mitigation contribution of manure management in the section “National deployment to meet the Chinese 1.5°C warming target”, and pointed out the emission of manure management in Fig. 5c. Thirdly, we clarified the positive effect on climate change by avoiding methane losses in the section “Discussion”. In that framework, we clarified the carbon mitigation contribution as fossil fuel substitution firstly, and depicted the methane control of manure in sequence to shown the overall cognition. Please check.

Biogas flow fitting and achieving carbon mitigation

Although the carbon mitigation potential of CBPD includes methane emission reduction of waste management and fossil fuel replacement, the contribution of manure management optimization was not calculated in this section, as the basic scenarios varied with different methane conversion factors, ranging from 0.1–80%. Therefore, we first calculated the carbon mitigation of fossil fuel replacement to clearly illustrate the improvement of an upgraded CBPD.

National deployment to meet the Chinese 1.5°C warming target

Apart from the contribution of energy replacement, we were also interested in the effect of Chinese national use of upgraded CBPDs (with CPR = 1) on methane emission mitigation of manure management, which is one of the main methane emission sources in rural areas. As there were still 186.7 million rural households in China in 2020, most of which had no access to natural gas, a national upgraded CBPD use could provide an alternative path to achieve energy equality and contribute 3.77% to meet the Chinese carbon mitigation commitment of the Paris Agreement of a 1.5°C increase in global warming (Fig. 5b), containing both methane emission control of manure and fossil fuel substitution.....

Discussion

.....and proposed that its broad application could have a vital positive effect on climate change mitigation, organic waste management optimization, and fossil fuel offsetting with the credits for energy supply on demand and avoiding biogas losses.

– Comment 3:

In the introduction, data about methane production potential is mentioned, but it's not entirely clear how this data was obtained or its reliability. It would be helpful to provide a brief explanation or source.

Response: (marked in red in the revised manuscript)

- Thanks very much for your kind notation. The methane production potential is sourced from the reference “NDRC & MOA. The 13th Five-Year Plan for biogas Development of the People's Republic of China. (NDRC & MOA, 2017)”, which is the newest publication literature from the government, and it is also similar to recent research, such as “China biogas gas industry. The China biogas industry development report: peaking carbon emissions 2030 and carbon neutrality 2060. (China biogas gas industry, 2023)”. We have added relative citations in the section “Introduction”. Please check.

– Comment 4:

The text in the introduction could benefit from a clearer, more focused problem statement at the beginning. It should explicitly state that the primary issue is the need for an efficient and widespread biogas distribution system that mitigates methane emissions.

Response: (marked in red in the revised manuscript)

- Thanks very much for the valuable suggestion. We have been stated at the first paragraph in the section “Introduction”. Firstly, the efficient and widespread biogas distribution system could be a viable negative emission path to replace fossil fuels, based on the waste-to-energy-produced pattern for methane control and fossil fuels replacement. Secondly, the broad use is significant to align the socio-economic development and climate objectives for sustainable development, Please check.

Introduction

Under many scenarios, fossil fuels will be the dominant energy source due to the lack of reliable and convenient renewable energy supply systems until 2050, but their growing use is causing increased carbon emissions. To alter this trend, biogas (55%–65% methane content) utilization, based on the waste-to-energy-produced pattern for methane control and fossil fuels replacement, is considered to be a viable negative emission path.

Thus, aligning the socio-economic development and climate objectives requires a wide adoption of efficient biogas production and distribution systems, to simultaneously and substantially decrease fossil fuel usage and mitigate methane emissions from organic matter degradation.

– Comment 5:

CBPD optimization for demand-dependent biogas production and distribution: If any specific models or equations are used in this section, it may be helpful to reference them for readers who want to explore them further.

Response: (marked in red in the revised manuscript)

- Thanks very much for the valuable suggestion. To further explicit the biogas flow, the main revision may be as follows. First, the principle how to establish the general model and the reference of biogas consumption rate (reference 7) is depicted in the section “Optimization of CBPD with demand-dependent biogas production”. Second, the model of biogas consumption rate is added in the section “Biogas flow and biogas storage capacity”, which is defined as the cumulative value of all types of customers. Third, the biogas storage design for zero emission is equal to the safety factor multiplied by the maximal value of residues biogas held in the biogas storage facility during the period of the feeding interval, which is also described in section “Biogas flow and biogas storage capacity”. Please check.

Optimization of CBPD with demand-dependent biogas production

..... overall biogas consumption could be predicted to increase with time, both temporally and spatially. Thus, a forward prediction was necessary to design a flexible production system.

The raw biogas production curve could be determined using variables related to the reactor configuration and operational parameters, and existing mathematical models or improved equations derived from related operational

data fitting ⁷. Sensitivity analyses and refinement of final biogas production curve, modified for amplitude and duration, would be carried out according to the actual data-driven modeling and step 4,.....

Biogas flow and biogas storage capacity

The biogas consumption rate is the superimposed effect of all customers' values, as shown in Equation 2.

$$\int_t^{t+\Delta t} g(t) = \sum_i g(t)_i \cdot N_{customer,i} \quad (2)$$

Where: $g(t)_i$ is average biogas consumption rate of customer type i in the region; $N_{customer,i}$ is the number of customer type i in the region.

The biogas storage capacity design would account for the sum of maximal storage capacity and margin design requirement, as shown in Equation 3.

$$Q_{storage} = k \times \text{Max}(Q(t):Q(t+\text{feeding interval})) \quad (3)$$

Where: $Q_{storage}$ is biogas storage capacity; k is safety factor for margin design to avoid biogas emission; $\text{Max}(Q(t):Q(t+\text{feeding interval}))$ is the maximal value of residual biogas held in biogas storage facility during the period of the feeding interval.

– Comment 6:

National Deployment to Meet the Chinese 1.5°C Warming Target: This section highlights the potential of decentralized upgraded CBPD to utilize animal waste as a readily available feedstock material. It effectively addresses the importance of local availability and its impact on broad CBPD deployment. However, the text could benefit from a more in-depth exploration of two vital aspects. Firstly, it would be valuable to delve into the challenges and solutions related to slurry management, a critical factor that can influence rural communities' willingness to adopt biogas systems. Effective strategies for efficient slurry management are key to mitigating one of the primary challenges faced by potential users.

Secondly, the text mentions that most of the Chinese mainland areas, with their high rates of methane production from manure and high domestic gas demand in the rural community (RPD), are likely candidates for the broad deployment of upgraded CBPD. Did the authors consider the changing landscape of rural areas, with livestock production moving closer to cities due to transportation cost considerations? This raises important questions about the adaptability and reach of biogas systems in evolving rural communities, particularly those that may no longer be considered purely rural areas. Further discussion on these aspects in the "Discussion and Policy Implications" section would provide a more comprehensive understanding of the biogas landscape's dynamics in rural China.

Response: (marked in red in the revised manuscript)

- Thanks very much for your valuable suggestion. The related revision has been done accordingly. First, the challenges related to slurry management mainly include spatial and

temporal heterogeneity of different manure sources, long distances of feedstock transportation, revenues of the feedstock supply chain, and so on. We have discussed these issues, and recommend solutions including co-fermentation (taking Shanghai as an example), region- and practice-specific feedstock collecting strategy, and clarifying the coordinated scenarios design for practical operation. These have been added in the section “National deployment to meet the Chinese 1.5°C warming target”. Second, we also have discussed the solution to the changing landscape of rural areas in the section "Discussion ", including co-fermentation, advanced algorithms, data analysis techniques, feedstock collecting plan, structural adjustment of animal husbandry and manure management. These could be used realize to fulfill biogas consumption on demand. Please check.

National deployment to meet the Chinese 1.5°C warming target

..... and co-fermentation with another organic waste is the potential path to fulfill biogas consumption. Even so, quantifying the characteristics and recovery percentages of different manure sources are still important task, due to the large spatial and temporal heterogeneity in estimating the associated biogas production potential. Furthermore, it also should be noted that long distances of feedstock transportation could possibly decrease the carbon mitigation contribution, owing to a large amount of fossil fuel use. The last but not the least, feedstock logistics would also be influenced by stakeholders’ revenues, transportation costs, and public interest for various rural communities, because the feedstock supply chain is a market interpretation. Thus, it is highly essential to identify the region- and practice-specific feedstock collecting strategy before the on-site CBPD deployments, including exploring specific biogas production rate models of segregated feedstock for co-fermentation with other kinds of agriculture waste, and properly organizing feedstock logistics to attain full-size biogas production. In addition, it is also important to clarify the coordinated scenarios for different feedstocks feeding

Discussion

Because spatial-temporal differences of feedstock availability and their characteristics are complicated and variable to realize efficient production and form beneficial products, advanced algorithms and data analysis techniques are efficient pathways to create positive impacts on quantifying biogas production variance (time requirement) and optimizing operation strategies in realistic cases. A specific feedstock collecting plan should be identified,

In addition, we highly recommend a structural adjustment of animal husbandry and manure management for high qualified feedstock collection and optimal regional logistics, in order to fully attain maximum energy and climate benefits.....

– Comment7:

Discussion and policy implications: While this discussion outlines the promising pathway for upgrading CBPD systems and achieving energy and climate benefits, it's important to acknowledge the challenges related to biogas emissions and the technical expertise required for successful implementation. Addressing the availability of skilled technicians and tackling biogas emission complexities in the context of rural communities should be a critical consideration in

further research and policy development. These challenges are pivotal in ensuring the practical feasibility and success of the proposed strategies.

Response: (marked in red in the revised manuscript)

- Thanks very much for your valuable suggestion. We have clarified the signification of biogas supply on demand and policy options to address the issues of skilled technician availability in the section “Discussion” firstly. To overcome the biogas emission in complexities, we pointed out that the solution of coordinated scenarios in the section “biogas flow fitting and achieving carbon mitigation”, and the potential strategy to adjust biogas usage dynamically by kinetic parameters learning in the scaled application, except for upgraded CBPD design and skillful manipulation, in the section “Discussion”. Please check.

Discussion

.....as the biogas supply on demand is significantly correlated with time dependency and transportation limits of organic waste management.

Such policy options must be evaluated in more detail to convince investors that upgrading CBPD is a reliable technology, as well as increase targeted subsidies for a combination of storage capacity optimization and operational training.

Biogas flow fitting and achieving carbon mitigation

it is highly recommended to establish an extra design of storage capacity in order to address another problem, defined as coordinated scenarios (Fig. 3b), such as adding an agreed upon threshold of a successional feedings interval of biogas production in order to accurately design the standing biogas supply process without error cumulating.....

Discussion

Except for upgraded CBPD design and skillful manipulation, a potential strategy leading to a more resilient system is to dynamically adjust biogas usage by kinetic parameters learned from detailed applications. Thus, further research should focus on auxiliary measures to minimize biogas storage capacity requirements and the negative impact of undesirable processes.....

Reviewer #2: (marked in blue for color highlighting in manuscript)

– Comment 1:

The authors explored the potential of an upgraded Community Biogas Production and Distribution system (CBPD) to meet households' cooking energy needs and mitigate greenhouse gas (GHG) emissions as an alternative to natural gas. Anaerobic digestion (AD) technology holds significant promise for providing clean energy for cooking, heating, and lighting, while simultaneously addressing indoor air pollution and organic waste management issues in rural areas. To fully realize its potential, widespread adoption of AD technology in rural settings is indispensable. The extensive uptake of AD technology as an energy producer hinges on ensuring both the reliable supply of biogas and the environmental and economic benefits of biogas as an energy source. In this context, the present study underscores the potential of the upgraded CBPD to ensure a demand-driven biogas supply while minimizing GHG emissions resulting from biogas slippage.

Response: (marked in blue in the revised manuscript)

- Thank you so much for your positive comment.

– Comment 2:

General comments:

The methodology section could benefit from greater detail. The transparency of the developed model and data analysis could be enhanced if the authors shared the collected data and the specifics of the developed model.

Response: (marked in blue in the revised manuscript)

- Thank you so much for your comment. The collected data and the specifics of the developed model were shared accordingly in the section “Data availability”, which have been uploaded by using the suggested figshare integration. More calculating models have been added in the section “Methods” as the equations 2, 3, 5, 7, 9, and 10. Please check.

– Comment 3:

The discussion presented in this study is solely derived from the current investigation. The authors did not undertake a comparative analysis or discussion of results in the context of related studies. While a direct comparison is not feasible due to the absence of a specific study, the authors could have drawn insights from different research endeavors exploring temporal and

spatial variations in feedstock availability for biogas production, the potential for biogas production, current and projected household energy demands, and the greenhouse gas mitigation potential of biogas.

Response: (marked in blue in the revised manuscript)

- Thank you so much for your valuable suggestion. We have done comparison with the full biogas usage systems, known as the Combined Heat and Power (CHP) unit and biomethane upgrading system in the section “Introduction”, the results indicate that the proposed CBPD would provide dominant advantages over conventional pattern, because it doesn’t need the traditional energy supply chain of exploitation, conversion, and distribution. In addition, we have mentioned the advantages of upgraded CBPD in lower middle-income and low-income countries to illustrate the feedstock availability for biogas production, the potential for biogas production on demand, and the potential of greenhouse gas mitigation in the section “National deployment to meet the Chinese 1.5°C warming target”, detail comparison is shown in Supplementary Table 4. The current and projected households’ energy demands (the huge rural energy demand and its clean renovation requirement of substituted solid fuel) have been discussed in the section “Discussion”. The greenhouse gas mitigation potential has been added in the section “National deployment to meet the Chinese 1.5°C warming target”, includes the national deployment current on current rural households’ biogas usage level, and further biogas usage with more patterns in rural communities. Please check.

Introduction

In the European Union, biogas is mainly utilized in either a Combined Heat and Power (CHP) unit or in a biomethane upgrading system, and the processed energies are delivered to electricity grids or gas distribution networks in a timely manner with guaranteed remuneration. These biogas systems require high quality monitoring devices and automatically controlled processes to minimize methane slipping. Its wide use in developing areas could face various challenges of economic feasibility and high level operational requirements, owing to both inadequate subsidies and insufficient technical support. On-site generation and direct supply of biogas to consumers would result in significant advantages to the traditional supply chain of exploitation, conversion, and distribution.

National deployment to meet the Chinese 1.5°C warming target

Although diverting organic waste through anaerobic digestion is the most effective approach toward net-zero warming among the mainstream technologies, a feasible system in lower middle-income and low-income countries should be simple and practical, which mainly include a family size household digester, CBPD, upgraded CBPD, and so on. After comparison and analysis (see Supplementary Table 4), upgraded CBPD could have the highest potential to efficiently utilize biogas and mitigate carbon for board application in current Chinese rural area.

Discussion

The methane production potential of manure in China is 2,467.7 billion MJ yr⁻¹, which only covers 26.3% of current direct residential consumption in rural areas. In addition, rural manufacturing industries would provide an exponential of energy consumption because of China's rural revitalization strategy. The huge rural energy demand and its clean energy renovation of solid fuel substituting also requires more upgraded CBPD deployment for converting various agricultural wastes to fulfill the time-use demand.

National deployment to meet the Chinese 1.5°C warming target

As Fig. 5c shows, current rural households' biogas usage only can consume 25.4% of manure's methane production potential at the investigated biogas consumption rate, as described in this study. An industrial or scalable level of biogas use is essential to supply more clean energy in rural communities, by exploring diverse kinds of energy consumption and adjusting the specific biogas flows. Although the direct electricity supply has the lowest environmental performance due to the low energy efficiency in converting biogas to electricity (see Supplementary Table 3), it still has a remarkably positive effect on climate change. Even if considering the contribution of fossil fuel substitution only, further biogas usage can have an extra GHG abatement of national fossil fuel combustion, ranging from 0.9–1.7%, regardless of the methane emission decline of manure management optimization.

– Comment 4:

The study calculates biogas production potential based on the total livestock number and their manure production potential. However, it is crucial to note that not all animal manure is necessarily available for biogas production. The recovery percentage of manure varies depending on factors such as animal types, location, and livestock production systems. It remains unclear whether the authors have factored in the recovery percentage of animal manure in their calculations of biogas production potential. Clarification on this aspect would strengthen the validity and applicability of the findings.

Response: (marked in blue in the revised manuscript)

- Thank you so much for your meaningful suggestion. Because the percentage and quality of manure is essential for biogas production, relative clarification has been added in corresponding parts, and we have discussed the solution strategies, and recommended that co-fermentation is the alternative path to fulfill biogas consumption once it is difficult to collect enough manure. Firstly, how to identify the region- and practice-specific feedstock collecting strategy have been added in the section "National deployment to meet the Chinese 1.5 °C warming target". Secondly, we also clarify that advanced analysis techniques and specific feedstock collecting plan are vital to further improve the operation performance, which have been added in section "Discussion". Please check.

National deployment to meet the Chinese 1.5°C warming target

.....as the rates of methane production potential from manure and domestic biogas demand in the rural regions (RPD) are fairly high in most areas. Shanghai is the only province, whose RPD is less than 1, and co-fermentation with another organic waste is the potential path to fulfill biogas consumption.

Even so, quantifying the characteristics and recovery percentages of different manure sources are still important task, due to the large spatial and temporal heterogeneity in estimating the associated biogas production potential. Furthermore, it also should be noted that long distances of feedstock transportation could possibly decrease the carbon mitigation contribution, owing to a large amount of fossil fuel use. The last but not the least, feedstock logistics would also be influenced by stakeholders' revenues, transportation costs, and public interest for various rural communities, because the feedstock supply chain is a market interpretation. Thus, it is highly essential to identify the region- and practice-specific feedstock collecting strategy before the on-site CBPD deployments, including exploring specific biogas production rate models of segregated feedstock for co-fermentation with other kinds of agriculture waste, and properly organizing feedstock logistics to attain full-size biogas production.

Discussion

Because feedstock availability and their characteristics are complicated and variable, owing to spatial-temporal differences, advanced algorithms and data analysis techniques are efficient pathways to create more positive impacts on quantifying biogas production variance (time requirement) and optimizing operation strategies to form beneficial products in realistic cases. A specific feedstock collecting plan should be identified, as the biogas supply on demand is significantly correlated with time dependency and transportation limits of organic waste management.

– Comment 5:

Crop residues have also been considered as a potential feedstock for biogas production. Crop residues are indeed a significant potential feedstock for biogas production, and it's crucial to consider their major types, availability, and biogas production potential. The data collection and analysis section of the paper, however, does not address these aspects. Moreover, crop residues not only differ in their biogas production potentials but also in their lag phase and production rates. A more detailed analysis would have significantly benefited the paper.

Response: (marked in blue in the revised manuscript)

- Thank you very much for your useful comment. We have shown the possible scenarios of different fermentation conditions, co-fermentation with crop residues is possible state; and clarify that the addressed strategy for biogas flow interactions is to add an allowable fluctuation in the section “Biogas flow fitting and achieving carbon mitigation”, which could solve the issues of the differentiated biogas production potential and lag phase (Fig. 3b). Further, how to collect the feedstock (identifying the region- and practice-specific feedstock collecting strategy), including the specific biogas production rate models of segregated feedstock for co-fermentation, have been depicted in section “National deployment to meet the Chinese 1.5°C warming target”. Finally, we also have clarified that advanced algorithms and data analysis techniques are vital for performance improvement, and a more resilient system could be established by kinetic parameters

learned from scaled applications, which are added in the section “Discussion”. Please check.

Biogas flow fitting and achieving carbon mitigation

Furthermore, it is highly recommended to establish an extra design of storage capacity in order to address another problem, defined as coordinated scenarios (Fig. 3b), such as adding an agreed upon threshold of a successional feedings interval of biogas production in order to accurately design the standing biogas supply process without error cumulating, because CPR is not always equal to 1 at one feeding interval. This means feeding parameters should be adjusted (feeding point or amount change) or maintained (a default operating process) for dynamic coordinating, which could be established on the basis of continuous information analyses of biogas consumption changes, residue biogas held in storage facilities, and biogas production variety during the last feeding interval.

Figure 3. Biogas storage design for synergistic adjustments. a. Variation of biogas held in storage facilities, with the given data shown during one feeding interval. The pink line is the ideal scenario, with its variation of biogas held in the storage facility based on biogas production and consumption curve fitting; colored points show the conflicted scenarios of the selected five biogas production curves to fulfill biogas consumption under the most complex situations (the highest R2 of the average biogas production of 5 continuous days). The total amount of biogas produced is assumed to equal that of biogas consumption during the feeding interval, and biogas held in the storage facility is the same at each feeding point.

National deployment to meet the Chinese 1.5°C warming target

Thus, it is highly essential to identify the region- and practice-specific feedstock collecting strategy before the on-site CBPD deployments, including exploring specific biogas production rate models of segregated feedstock for co-fermentation with other kinds of agriculture waste, and properly organizing feedstock logistics to attain full-size biogas production. In addition, it is also important to clarify the coordinated scenarios for different feedstocks feeding.

Discussion

Because feedstock availability and their characteristics are complicated and variable, owing to spatial-temporal differences, advanced algorithms and data analysis techniques are efficient pathways to create more positive impacts on quantifying biogas production variance (time requirement).....

.....a potential strategy leading to a more resilient system is to dynamically adjust biogas usage by kinetic parameters learned from detailed applications.

– Comment 6:

The authors have focused on calculating the greenhouse gas emissions mitigation potential of biogas as a substitute for natural gas in meeting rural households' energy demand for cooking. However, the paper acknowledges that natural gas is not readily available in all rural areas. Rural areas without easy access to clean energy options commonly rely on conventional sources such as firewood, crop residues, and dried animal dung for their cooking and heating needs. Therefore,

calculating greenhouse gas emissions mitigation potential based solely on natural gas substitutions may not fully reflect the actual energy practices in rural areas.

Response: (marked in blue in the revised manuscript)

- Thank you so much for your valuable comment. That is true that the structure of rural residential energy use varies, we have recalculated the greenhouse gas emissions mitigation potential with the united emission of rural residential energy weighted mean value of 0.0739 kg CO₂ eq /MJ, which is based on Chinese rural residential energy types and their percentages. The detail determination process could be found in the references “Li, J., Chen, C. & Liu, H. Transition from non-commercial to commercial energy in rural China: Insights from the accessibility and affordability. *Energy Policy* **127**, 392-403 (2019).”, and “MEE. Guidelines of action plans preparation in provincial level towards peaking carbon dioxide emissions. (MEE, 2021)”. Please check in the section “GHG emission calculations for the CBPD” and Supplementary Table 6.

GHG emission calculations for the CBPD

C_j is the fraction of biogas used to substitute energy type j ; E_j is the energy-specific CO₂ emission factor of energy type j use (see Supplementary Table 6), the carbon emission value of rural households’ energy use in this study was 0.0739 kg CO₂ eq MJ⁻¹, which is equal to the united emission of rural residential energy weighted mean value, calculated by the energy structure and relative emission factors.

– Comment 7:

The calculations in the study are based on the current household energy demand for cooking and the biogas production potential of animal manure. However, it is acknowledged that both the energy demand for cooking and the amount of animal manure available for biogas production are subject to change over time. It would be valuable for the developed model to incorporate sensitivity analysis to project the greenhouse gas mitigation potential of the CBPD under different scenarios, accounting for potential variations in energy demand and feedstock availability over time.

Response: (marked in blue in the revised manuscript)

- Thank you so much for your valuable suggestion. The sensitivity analysis on different scenarios for its change over time has been carried out in the section "Biogas flow fitting and achieving carbon mitigation" and Fig. 3a. The results show that 1.79 times biogas storage capacity amplification is the possible solution to adjust the most conflicted scenarios of the five investigated cases with model analysis, and method for further coordinated scenarios design could be found in Fig. 3b. The variations in feedstock availability over time could be solved with the region- and practice-specific feedstock

collecting strategy to attain full-size biogas production, which could be clarified in the section "National deployment to meet the Chinese 1.5°C warming target". The CBPD under different scenarios for the greenhouse gas mitigation potential have been analysis in the section "National deployment to meet the Chinese 1.5°C warming target", not only for rural households' biogas usage but also for the further biogas usage pathways (Fig. 5c). Please check.

Biogas flow fitting and its carbon mitigation achieving

With sensitive analyses using the established data, 1.79 times biogas storage capacity amplification would realize the robustness and practicality of the proposed CBPD (Fig. 3a), which can avoid biogas losses in the most conflicted scenarios. Furthermore, it is highly recommended to establish an extra design of storage capacity in order to address another problem, defined as coordinated scenarios (Fig. 3b), such as adding an agreed upon threshold of a successional feedings interval of biogas production in order to accurately design the standing biogas supply process without error cumulating, because CPR is not always equal to 1 at one feeding interval.

National deployment to meet the Chinese 1.5°C warming target

.....it is highly essential to identify the region- and practice-specific feedstock collecting strategy before the on-site CBPD deployments, including exploring specific biogas production rate models of segregated feedstock for co-fermentation with other kinds of agriculture waste, and properly organizing feedstock logistics to attain full-size biogas production.

As Fig. 5c shows, current rural households' biogas usage only can consume 25.4% of manure's methane production potential at the investigated biogas consumption rate, as described in this study. An industrial or scalable level of biogas use is essential to supply more clean energy in rural communities, by exploring diverse kinds of energy consumption and adjusting the specific biogas flows. Although the direct electricity supply has the lowest environmental performance due to the low energy efficiency in converting biogas to electricity (see Supplementary Table 3), it still has a remarkably positive effect on climate change. Even if considering the contribution of fossil fuel substitution only, further biogas usage can have an extra GHG abatement of national fossil fuel combustion, ranging from 0.9–1.7%, regardless of the methane emission decline of manure management optimization.

– Comment 8:

Furthermore, the study maintains a constant digester volume of 100 m³ throughout the calculations. The rationale behind selecting this specific volume is not clearly explained. It's essential to clarify how the chosen digester volume aligns with the available feedstock (i.e., animal manure and crop residues) and total energy demand for cooking, which can vary across different rural settings. Providing a clear explanation for selecting a particular digester volume and illustrating how it accommodates variations in both feedstock supply and energy demand would enhance the manuscript.

Response: (marked in blue in the revised manuscript)

- Thank you so much for your comment. The digester volume of 100 m³ is a representative scale for Chinese rural communities' cooking usage; which could provide a possible

solution to supply biogas for rural concentrated settlements. Because the household number of rural settlements is around 100, which is assumed with a provision of 1 m³ biogas customer per day for rural households' cooking usage. Thus, we have added an explanation for selecting the specified digester volume accordingly, and clarified that a higher fermented volume could have a better performance due to a lower percentage of digester heating requirements for universal deployment. Please check in “Supplementary Note 3 The climate conditions of 10 cities and their detailed parameters”.

Supplementary Note 3 The climate conditions of 10 cities and their detailed parameters

The small-scale biogas plants are likely to be most feasible in Chinese rural areas with fermentation volume of approximate 100 m³, which could provide a possible solution to supply biogas for rural concentrated settlements, as the household number of rural settlement is around 100 m³ for cooking usage with a provision of 1 m³ of biogas per customer per day. In addition, the biogas production capacity could reach 100 m³ or be at a low operational level, which can cover the most scenarios of usage with zero slipping for flexible production, not only for cooking, but also for heating, and so on. Thus, we selected an ungraded community biogas production and distribution system of 100 m³ fermented volume as the specific volume to study the performance of carbon mitigation in this study. Certainly, a higher fermented volume could have a better performance due to a lower percentage of digester heating requirements for universal deployment.

– Comment 9:

Including sensitivity analysis in the study would be beneficial. For instance, assessing the sensitivity of the estimated greenhouse gas mitigation potential to factors such as the size of the digester, storage tank, or biogas demand would provide valuable insights. If the results are found to be less sensitive to the size of the biogas storage tank, considering a slightly larger tank may better ensure a stable biogas supply in the event of unintended sudden fluctuations in biogas production or demand. This consideration would contribute to the robustness and practicality of the proposed CBPD system.

Response: (marked in blue in the revised manuscript)

- Thank you so much for your comment. The variation of biogas held in the biogas facility and the sensitivity analysis of biogas capacity requirement has been added according, as well as the synergistic adjustments of different scenarios, in the section “Biogas flow fitting and achieving carbon mitigation” and Fig. 3. It includes how to design biogas storage capacity to avoid biogas losses under the ideal scenarios, conflicted situation, and coordinated scenarios. Meanwhile, the potential strategy leading to a more resilient system and further research of auxiliary measures (to minimize biogas storage capacity requirements and impact of undesirable processes) have been depicted in the section “Discussion”. Please check.

Biogas flow fitting and the carbon mitigation potential

With sensitive analyses using the established data, 1.79 times biogas storage capacity amplification would realize the robustness and practicality of the proposed CBPD (Fig. 3a), which can avoid biogas losses in the most conflicted scenarios. Furthermore, it is highly recommended to establish an extra design of storage capacity in order to address another problem, defined as coordinated scenarios (Fig. 3b), such as adding an agreed upon threshold of a successional feedings interval of biogas production in order to accurately design the standing biogas supply process without error cumulating, because CPR is not always equal to 1 at one feeding interval. This means feeding parameters should be adjusted (feeding point or amount change) or maintained (a default operating process) for dynamic coordinating, which could be established on the basis of continuous information analyses of biogas consumption changes, residue biogas held in storage facilities, and biogas production variety during the last feeding interval.

Figure 3. Biogas storage design for synergistic adjustments.

Discussion

Except for upgraded CBPD design and skillful manipulation, a potential strategy leading to a more resilient system is to dynamically adjust biogas usage by kinetic parameters learned from detailed applications. Thus, further research should focus on auxiliary measures to minimize biogas storage capacity requirements and the negative impact of undesirable processes (for example, the temporal congestion caused by inaccurate feeding time points or amount selections without intervention), which contain the policy of time-of-use pricing at peaks and trough biogas usage, accurate predicting further scenarios of energy usage, and flexible biogas supply strategy to manufacturing industries

– Comment 10:

Specific comments:

The manuscript requires thorough revision for both English language usage and technical terminologies.

Response: (marked in blue in the revised manuscript)

- Thank you very much for your comment. We have invited the language polishing agency “International Science Editing (<http://www.internationalscienceediting.com>)” to help this manuscript edition, meanwhile, we also has been checked and revised as instructed several times. Please check.

– Comment 11:

Line 128: Should it be "1 m³ of biogas per customer per day"?

Response: (marked in blue in the revised manuscript)

- Thanks very much for your careful check. We have revised it according. Please check.

– Comment 12:

Lines 145 & 146: Please provide the unit for GHG emission values.

Response: (marked in blue in the revised manuscript)

- Thanks very much for pointing this out. We have revised it as "kg CO₂eq d⁻¹ customer⁻¹" accordingly. Please check.

– Comment 13:

Line 193: Please define RPD.

Response: (marked in blue in the revised manuscript)

- Thanks very much for your kind suggestion. RPD had been defined as “the rates of methane production potential from manure and domestic biogas demand in the rural regions” in the section “National deployment to meet the Chinese 1.5°C warming target”, and the calculation method is expressed as “the methane production potential of available manure divided by the total rural biogas demand in the certain area” in the section “RPD”. Please check.

National deployment to meet the Chinese 1.5°C warming target

.....as the rates of methane production potential from manure and domestic biogas demand in the rural regions (RPD).

RPD

.....the methane production potential of available manure divided by the total rural biogas demand in the certain area, as shown in Equation 6.

– Comment 14:

Lines 210-241: Further elaboration and clarification are needed. In its current form, it is unclear whether the authors are presenting their own findings or stating literature values.

Response: (marked in blue in the revised manuscript)

- Thanks very much for your reminding. Our finding is the contribution of the national upgraded CBPD deployment (contribute 3.77% to the Chinese carbon mitigation commitment), the referenced literature is used to illustrate the calculation basis (calculated using the recent Chinese report submitted to the United Nations). We have revised the two parts. Please check in the section “National deployment to meet the Chinese 1.5°C warming target”.

National deployment to meet Chinese 1.5°C warming target

.....a national upgraded CBPD use could provide an alternative path to achieve energy equality and contribute 3.77% to meet the Chinese carbon mitigation commitment of the Paris Agreement of a 1.5°C increase in global warming (Fig. 5b), containing both methane emission control of manure and fossil fuel substitution, which was calculated by using the recent Chinese report submitted to the United Nations.

– Comment 15:

Lines 231-236: In rural areas, animal manure is typically applied to agricultural land as Farmyard Manure (FYM). This implies that even in the absence of the proposed CBPD, animal manures will find their ultimate fate in agricultural land. Therefore, the authors may need to explain the added merits of digestate over FYM rather than simply overemphasizing the environmental merits of digestate.

Response: (marked in blue in the revised manuscript)

- Thanks very much for your valuable comment. We have converted the discussion on digestate over chemical fertilizers to that of effective management of manure with the proposed CBPD. The benefits of the ecological utilization mainly include the high feasibility of digested fertilizer utilization, decreasing direct nitrous oxide emissions, and recycling more manure nitrogen back to farmland instead of synthetic fertilizer. Please check in section “Discussion”.

Discussion

CBPD’s deployment would be carried out in a decentralized manner of surrounding rural communities, which can store and supply organic fertilizer on-site to minimize distribution distances, decrease direct nitrous oxide emissions by avoiding denitrification processes, and reduce synthetic fertilizer inputs because more manure nitrogen is recycled back to farmland instead of being released into the open air. These could facilitate ecological utilization of an anaerobically digested fertilizer to nearby farms.....

– Comment 16:

Line 294: Please check for "RPD" and "PRD."

Response: (marked in blue in the revised manuscript)

- Thanks very much for pointing this out. PRD is a worry, and we have revised it as RPD accordingly. Please check.

– Comment 17:

In Supplementary Table 1, the authors have mentioned CPR values of less than 80% for all the rural areas studied. Since households are using natural gas and firewood to meet their cooking energy demand, are such CPR values less than 1 due to a disparity in the production and consumption of biogas or due to the higher cost of biogas compared to firewood or the convenience of using natural gas?

Response: (marked in blue in the revised manuscript)

- Thanks very much for your comment. The reasons why biogas is thought as the practical selection for Chinese rural resident energy sources are clarified as followings. Firstly, the disparity in rates of production and consumption means that some parts of biogas are slipping as the inefficient biogas production or storage, which would result in a higher requirement of feedstock amount and operation costs for united biogas production. Currently, the biogas price is still lower than that of petrol gas, which indicates the biogas' price is competitive; and we have revised the description according in the section "Supplementary Table 1". Further, we have pointed out that broad application of natural gas facilities doesn't seem a feasible pathway for clean energy providing owing to the insufficient provision of natural gas, while biogas is the possible way; these are depicted in the section "Introduction". Thirdly, we also show the huge rural energy demand and its clean renovation requirement, which needs more upgraded CBPD deployment for converting various agricultural wastes to fulfill the time-use demand in current and future states; These are clarified in the section "Discussion". Please check.

Supplementary Table 1

From a least-cost perspective, the direct use of biogas as fuel seemed more reasonable than petrol gas, as the stability of biogas supply met the customers' requirement of energy service, and the price of biogas use was 0.084 Chinese Yuan MJ⁻¹ compared with approximately 0.112 Chinese Yuan MJ⁻¹ for petrol gas.

Introduction

Extensive development of natural gas facilities is not a feasible option, which would cause serious shortages during peak periods and significantly threaten energy security, such as the frequent gas shortage of coal-to-gas policies implemented in northern China. Biogas is therefore considered an essential and efficient technology to produce sustainable energy.....

Discussion

In addition, rural manufacturing industries would provide an exponential of energy consumption because of China's rural revitalization strategy. The huge rural energy demand and its clean energy renovation of solid fuel substituting also requires more upgraded CBPD deployment for converting various agricultural wastes to fulfill the time-use demand.

– Comment 18:

In Supplementary Table 2, please specify whether the OLR value is based on a wet weight or dry weight basis.

Response: (marked in blue in the revised manuscript)

- Thanks very much for your kindly comment. The OLR value is based on a wet weight, and we have revised as “Average OLR, kg wet weighted material d⁻¹” in “Supplementary Table 2”. Please check.

– Comment 19:

In Supplementary Figure 1, please explain the meaning of "biogas residue."

Response: (marked in blue in the revised manuscript)

- Thanks very much for your comment. It means the minimum requirement of residue biogas held in biogas storage facility at the feeding time points, which is used to efficiently replenish biogas production shortages during the feeding interval, normally at the beginning phase. We have redefined and revised as “Residue biogas” in “Supplementary Note 1. Effect of feeding time point on requirements of storage capacity and residue biogas held in a storage facility at the substrate loading site”. Please check.

Supplementary Note 1. Effect of feeding time point on requirements of storage capacity and residue biogas held in a storage facility at the substrate loading site

Appropriate biogas storage capacity with sufficient biogas presetting (residue biogas) is the necessary condition for an ensured biogas supply. The requirements for storage capacity and its minimum residue biogas at different feeding time points are shown in Supplementary Fig. 1.

Supplementary Figure 1. Requirements of storage capacity and its minimum residue biogas at different feeding time points

Residue biogas amount

Reviewer #3: (marked in green for color highlighting in manuscript)

– Comment 1:

The manuscript Titled “Unlocking the full potential of biogas systems for sustainable energy production and climate solutions in rural communities” presents a valuable contribution to the field of sustainable energy, particularly in the context of rural biogas systems. The proposed CBPD system has the potential to make a significant impact in terms of energy equality and carbon emissions reduction. However, the study would benefit from a more in-depth analysis of the methodologies, a critical evaluation of its limitations, and a more direct connection between the findings and policy implications.

Response: (marked in green in the revised manuscript)

- Thank you so much for your valuable comment. A more in-depth analysis of the methodologies has been added and revised mainly in the section “Biogas flow and biogas storage capacity”, “GHG emission calculations for the CBPD” and “Methane mitigation calculations of manure management optimization” as the equations 2, 3, 4, 5, 7, 8, 9, and 10. The limitations of the study mainly contain lacking advanced algorithms and data analysis techniques in realistic case and RPD analysis only conducting at the provincial scale for roughly calculation; relative discussion and optimizing strategies have been clarified in the section “Discussion”. The connection between the findings and policy implications has been done accordingly in the section “Discussion” according to each paragraph discussion, which mainly include acquisition with the perception of upgraded CBPD, increasing targeted subsidies for a combination of storage capacity optimization and operational training, structural adjustment of feedstock collection and optimal regional logistics, all elements integration for a more intrinsic synergistic interaction, auxiliary measures to minimize biogas storage capacity requirements and effect of undesirable processes, designing administrative regulation to cultivate professional organizations to provide reliable biogas supply service, and combinations of upgraded CBPD and other renewable energy systems. Please check.

Biogas flow and biogas storage capacity

The biogas consumption rate is the superimposed effect of all customers' values, as shown in Equation 2.

The biogas storage capacity design would account for the sum of maximal storage capacity and margin design requirement, as shown in Equation 3.

GHG emission calculations for the CBPD

In this study, GHG emissions of CBPD at a time interval of Δt (E_{CBPD} , kg CO₂eq d⁻¹ customer⁻¹) only accounted for utilizable biogas and methane slipping due to biogas emissions into the open air, as shown in Equation 4.

Net GHG emissions of upgraded CBPD on site at time interval of Δt ($E_{\text{CBPD, net}}$, kg CO₂eq d⁻¹ customer⁻¹ accounted for net biogas production, as shown in Equation 5.

Methane mitigation calculations of manure management optimization

The amount of methane emission from manure management in the region (CH₄ emission) is expressed as in Equation 7.

Calculated carbon mitigation relative to the baseline scenario (CMB) of national deployment and its contribution for meeting the Chinese 1.5°C target ($RT_{1.5^\circ\text{C}}$) is expressed as in Equation 8 and Equation 9.

Carbon contribution of fossil fuel replacement (E_{replace}) is expressed as in Equation 10.

Discussion

..... advanced algorithms and data analysis techniques are efficient pathways to create more positive impacts on quantifying biogas production variance (time requirement) and optimizing operation strategies to form beneficial products in realistic cases.

A limitation of this study was that the RPD analysis was conducted at the provincial scale. Once it is investigated at small regional levels, such as towns, a more accurate targeted result could be obtained.

Such policy options must be evaluated in more detail to convince investors that upgrading CBPD is a reliable technology, as well as increase targeted subsidies for a combination of storage capacity optimization and operational training.

..... we highly recommend a structural adjustment of regional animal husbandry and manure management improvements for high qualified feedstock collection, as current amount of poultry and livestock farming is increasing and manure management strategies are varied.

These could facilitate ecological utilization of an anaerobically digested fertilizer to nearby farms, which means that once all processes can be regionally integrated well, such as biogas consumption, breeding scale, and planting structures, and further benefits of circular agriculture realization can be achieved with the intrinsic synergistic interaction. Thus, rural communities in developing areas are encouraged to deploy upgraded CBPDs to achieve energy equality and regional carbon mitigation.....

.....further research should focus on auxiliary measures to minimize biogas storage capacity requirements and the negative impact of undesirable processes.....

Further policy measures should include designing administrative regulation to cultivate professional organizations to provide reliable biogas supply service, and effective judgment criteria for methane mitigation subsidies based on institutional designs using Common Pool Resource theory..... combinations of upgraded CBPD and other renewable energy systems, could develop superior synergistic systems for more stand-alone energy supplies with high primary energy use ratios by ingeniously applying the characteristics of flexible energy production.....

– Comment 2:

Abstract and Introduction

□ The abstract concisely outlines the objective and scope of the study, highlighting the development of an upgraded community biogas production and distribution system (CBPD) and its potential impact on carbon emissions reduction. However, it could benefit from a more detailed explanation of the methodologies and key findings

Response: (marked in green in the revised manuscript)

- Thanks very much for your suggestion. We have revised it as instructed. Methodologies include taking five current systems as empirical examples, mechanisms of synergistic adjusting, exploring with rural residential biogas supply, viability analysis, and so on. The key findings include the improved design of upgraded community biogas production and distribution system, the strategy to acquire consumption-to-production ratios of close to 1, the feasibility of deployment under a universal prevailing climate, the carbon mitigation contribution of Chinese national deployment, and decarbonization potential. Please check.

Abstract:

.....realize energy equality and mitigate climate change in a highly reliable manner; however, existing approaches usually ignore either full biogas supplies in dynamic situations or methane losses. Here, we propose community biogas production and distribution system and report an improved design to achieve full co-benefits in developing economies, taking five existing systems as empirical examples; mechanisms of synergistic adjusting out-of-step biogas flow rates on both plant- and user-sides are defined to obtain consumption-to-production ratios of close to 1, which are explored with rural residential biogas supply; carbon emissions reduction and its viability under universal prevailing climate are illustrated. Coupled with manure management optimization, Chinese national deployment of this rural residential system would contribute a 3.77% reduction towards meeting its global 1.5°C target. Furthermore, fulfilling communities' energy demand has a much greater decarbonization potential.

– Comment 3:

The introduction effectively sets the context for the study, emphasizing the significance of biogas as a sustainable energy source. However, a more comprehensive discussion of the current challenges in biogas system implementation, particularly in rural contexts, would strengthen the background

Response: (marked in green in the revised manuscript)

- Thank you so much for your valuable suggestion. The obstacles to biogas system implementation in developing rural areas mainly include intermittent biogas supply and low use rate of produced biogas products, inadequate subsidies and insufficient technical support, challenges of economic feasibility and high-level operational requirements, lacking the CBPD optimization and operation strategies to achieve the prerequisites of both biogas supply on demand and close-to-zero methane leakage in developing rural

area. These factors hinder the broad application, which also are the main research content in this study for energy equality and carbon emissions reduction. The background description has been added in the section “Introduction”. Please check.

Introduction

Despite biogas’s extensive history in cooking, heating, and power generation, including its use in biogas-based natural gas production, its contribution to the current energy mix and its climate benefits remain modest in rural developing areas. This is largely due to intermittent energy supply challenges and the underutilization of collected methane

Its wide use in developing areas could face various challenges of economic feasibility and high level operational requirements, owing to both inadequate subsidies and insufficient technical support. On-site generation and direct supply of biogas to consumers would result in significant advantages to the traditional supply chain of exploitation, conversion, and distribution. It could result in the best co-benefits of high quality services of clean energy providing and greenhouse gas (GHG) emission mitigation, including the prerequisites of on-demand biogas supply and close-to-zero methane leakage.

In this study, the development of a CBPD optimization framework and its operational strategies are keys to form a self-sufficient system capable of independent operation without external support, and provide innovative solutions that are more feasible and efficient than those found in conventional systems

– Comment 4:

Methodology

□ The methodology section provides a clear description of the steps involved in optimizing the CBPD system. It includes the identification of biogas demand rate, quantification of operational parameters, design of biogas storage capacity, and determination of operational strategy.

The data analysis process is well-detailed, especially in handling data inconsistencies and errors. However, the manuscript could benefit from a more in-depth discussion on the choice of methods and tools used for data collection and analysis, particularly the use of multipath ultrasonic gas flow meters.

Response: (marked in green in the revised manuscript)

- Thank you so much for your valuable remark. It is been revised as instructed. Firstly, the advantages of transit-time ultrasonic gas flow meters and equipment methods have been added. Secondly, the method of the dehydration system equipping is also illustrated to avoid dew formation. These are shown in the section "Data analysis of CBPD". Thirdly, the carbon emission value of rural households’ energy use in this study is shown in the section "GHG emission calculations for the CBPD". Fourthly, methods of $N_{household}$ and $S_{average}$ determination are given in the section "RPD". Similar revising or refining for data collection and analysis has been done as possible in some other place. Please check.

Data analysis of CBPD

Hourly biogas production, hourly biogas consumption, and methane content of the five selected CBPDs were measured using transit-time ultrasonic gas flow meters (TY1030, TianYu, Wuhan, China), which are easy to install with minimal or no disruptions to the flow and have several vital advantages, such as a small pressure drop, high accuracy and wide range ratio. Each CBPD is equipped two meters with dehydration system ahead, which is used to avoiding dew formation⁵³. One is before the biogas storage facility for production flow measuring; the other is on the main pipeline of biogas supply. All meters have been calibrated and validated on the test facilities for biogas in local institute of measurement & testing every half year.

GHG emission calculations for the CBPD

.....the carbon emission value of rural households' energy use in this study was 0.0739 kg CO₂ eq MJ⁻¹, which is equal to the united emission of rural residential energy weighted mean value, calculated by the energy structure and relative emission factors.

RPD

..... $N_{household}$ was obtained from a national population census, and $S_{average}$ was calculated from the data for the five CBPDs examined in this study.

– Comment 5:

Results and Discussion

□ The results section presents a comprehensive analysis of the biogas production and consumption patterns. However, there is a lack of detailed statistical analysis or comparative study to validate the effectiveness of the proposed CBPD optimization strategies.

Response: (marked in green in the revised manuscript)

- Thank you so much for your valuable remark. The detailed statistical analysis of the established data has been analyzed as instructed, which is used to illustrate the synergistic adjustments of the proposed CBPD to address the most coordinated scenarios. In addition, we also show the adjustment strategy to realize the stable biogas supply and slipping remove, as well as addressing the asynchronization of two successional times feeding. These are shown in the section “Biogas flow fitting and achieving carbon mitigation” and Figure 3. Besides, proposed CBPD has two advantages on the popular application (CHP system) for carbon mitigation, stable operation in the long term and general less transport distance, which are shown in the section “State-of-the-art of upgraded CBPD during prevailing climate”. Please check.

Biogas flow fitting and achieving carbon mitigation

With sensitive analyses using the established data, 1.79 times biogas storage capacity amplification would realize the robustness and practicality of the proposed CBPD (Fig. 3a), which can avoid biogas losses in the most conflicted scenarios. Furthermore, it is highly recommended to establish an extra design of storage capacity in order to address another problem, defined as coordinated scenarios (Fig. 3b), such as adding an

agreed upon threshold of a successional feedings interval of biogas production in order to accurately design the standing biogas supply process without error cumulating, because CPR is not always equal to 1 at one feeding interval. This means feeding parameters should be adjusted (feeding point or amount change) or maintained (a default operating process) for dynamic coordinating, which could be established on the basis of continuous information analyses of biogas consumption changes, residue biogas held in storage facilities, and biogas production variety during the last feeding interval.

Figure 3. Biogas storage design for synergistic adjustments.

State-of-the-art of upgraded CBPD during prevailing climate

Furthermore, it could have a greater feasibility of decarbonization contributions than that of CHP, which normally has a highly volatile operation revenues, depending on the real-time electricity prices for spatio-temporal variations and the general longer transport distance. After comprehensive consideration, upgraded CBPDs have the potential to be widely used as scalable decarbonization solutions in developing economics.

– Comment 6:

The discussion effectively highlights the potential impact of CBPD systems on energy equality and carbon emissions reduction in rural communities. However, it could be enhanced by including comparisons with existing biogas systems and discussing the scalability and practical challenges of implementing the proposed system.

Response: (marked in green in the revised manuscript)

- Thank you so much for your valuable remark. It is been revised as instructed. We have pointed out that the feasible systems in lower middle-income and low-income countries mainly include a family size household digester, CBPD, upgraded CBPD, and so on. Further, we have compared upgraded CBPD to existing biogas systems on main biogas usage pattern, practical challenge, methane emission, scalability, and prospects. The results show that upgraded CBPD has the highest potential to utilize biogas and mitigate carbon efficiently in rural areas. Please check in the section “National deployment to meet the Chinese 1.5°C warming target” and Supplementary Table 4.

National deployment to meet the Chinese 1.5°C warming target

Although diverting organic waste through anaerobic digestion is the most effective approach toward net-zero warming among the mainstream technologies, a feasible system in lower middle-income and low-income countries should be simple and practical, which mainly include a family size household digester, CBPD, upgraded CBPD, and so on. After comparison and analysis (see Supplementary Table 4), upgraded CBPD could have the highest potential to efficiently utilize biogas and mitigate carbon for board application in current Chinese rural area.

Supplementary Table 4 Comparisons of upgraded community biogas production and distribution system (CBPD) with existing biogas systems for implementation

– Comment 7:

Conclusions and Policy Implications

□ The manuscript concludes with significant findings, emphasizing the potential of CBPD systems in reducing carbon emissions and achieving energy equality in rural areas. However, the conclusions could be more impactful by summarizing key findings more explicitly and suggesting future research directions.

Response: (marked in green in the revised manuscript)

- Thank you so much for your valuable remark. Key findings have been revised as instructed, which mainly include acquiring the kinetic parameters associated with plant- and user-sides, designing flexibility of biogas flow interaction, establishing the coordination-adjusting mechanism, climate change mitigation at prevailing climate conditions, and credits for energy supply. Future research has been explicated accordingly, including auxiliary measures to minimize biogas storage capacity requirements, and superior synergistic systems for more stand-alone energy supplies by ingeniously using the characteristics of flexible energy production. Please check in section “Discussion”

Discussion

The upgraded CBPD could be formed by acquiring the kinetic parameters associated with plant- and user-sides, designing a flexible interaction, and establishing coordination-adjusting mechanisms for each successive feeding. We also showed that upgraded CBPD could produce sustainable energy at prevailing climate conditions, and proposed that its broad application could have a vital positive effect on climate change mitigation, organic waste management optimization, and fossil fuel offsetting with the credits for energy supply on demand and avoiding biogas losses.

.....further research should focus on auxiliary measures to minimize biogas storage capacity requirements and the negative impact of undesirable processes (for example, the temporal congestion caused by inaccurate feeding time points or amount selections without intervention), which contain the policy of time-of-use pricing at peaks and trough biogas usage, accurate predicting further scenarios of energy usage, and flexible biogas supply strategy to manufacturing industries.

Last but not least, combinations of upgraded CBPD and other renewable energy systems, such as solar energy, heat pumps, etc., could develop superior synergistic systems for more stand-alone energy supplies with high primary energy use ratios by ingeniously applying the characteristics of flexible energy production, which provides participants or regulators a way to identify more and better benefits.

– Comment 8:

The policy implications section is insightful, focusing on the need for improved managerial skills and the integration of CBPD with other renewable energy systems. A more explicit connection between the research findings and specific policy recommendations would be beneficial.

Response: (marked in green in the revised manuscript)

- Thank you so much for your kindly suggestion. The more explicit connection between the research findings and specific policy recommendations has been improved in each paragraph after the discussion on relative findings. The policy recommendations include acquisition with the perception of upgraded CBPD as well as increase targeted subsidies for a combination of storage capacity optimization and operational training, structural adjustment of feedstock collection and optimal regional logistics, all elements integration for a more intrinsic synergistic interaction, auxiliary measures to minimize biogas storage capacity requirements and effect of undesirable processes, designing administrative regulation to cultivate professional organizations to provide reliable biogas supply service, and establishing the combinations of upgraded CBPD and other renewable energy systems. Please check in section “Discussion”.

Discussion

Such policy options must be evaluated in more detail to convince investors that upgrading CBPD is a reliable technology, as well as increase targeted subsidies for a combination of storage capacity optimization and operational training.

..... we highly recommend a structural adjustment of regional animal husbandry and manure management improvements for high qualified feedstock collection, as current amount of poultry and livestock farming is increasing and manure management strategies are varied.

These could facilitate ecological utilization of an anaerobically digested fertilizer to nearby farms, which means that once all processes can be regionally integrated well, such as biogas consumption, breeding scale, and planting structures, and further benefits of circular agriculture realization can be achieved with the intrinsic synergistic interaction. Thus, rural communities in developing areas are encouraged to deploy upgraded CBPDs to achieve energy equality and regional carbon mitigation.....

.....further research should focus on auxiliary measures to minimize biogas storage capacity requirements and the negative impact of undesirable processes.....

Further policy measures should include designing administrative regulation to cultivate professional organizations to provide reliable biogas supply service, and effective judgment criteria for methane mitigation subsidies based on institutional designs using Common Pool Resource theory..... combinations of upgraded CBPD and other renewable energy systems, could develop superior synergistic systems for more stand-alone energy supplies with high primary energy use ratios by ingeniously applying the characteristics of flexible energy production.....

– Comment 9:

General Comments

□ The manuscript is well-structured and presents a novel approach to optimizing CBPD systems. However, it would benefit from a more critical analysis of the limitations of the study and potential barriers to implementation.

Response: (marked in green in the revised manuscript)

- Thank you so much for your valuable suggestion. The limitations of the study mainly contain data analysis techniques in realistic case and RPD analysis only conducting at the provincial scale for roughly calculation, and lacking advanced algorithms and data analysis techniques in realistic case. These limitation and relative solution have been clarified in the section “Discussion”. Potential barriers on implementation mainly include the region- and practice-specific feedstock collecting strategy, the perception of upgraded CBPD, targeted subsidies for a combination of storage capacity optimization and operational training, and so on. We have pointed out the current dilemma and discussed the solution strategies. Please check in sections “National deployment to meet the Chinese 1.5°C warming target” and “Discussion”.

Discussion

A limitation of this study was that the RPD analysis was conducted at the provincial scale. Once it is investigated at small regional levels, such as towns, a more accurate targeted result could be obtained. To attain maximum energy and climate benefits in practical, we highly recommend a structural adjustment of regional animal husbandry and manure management improvements for high qualified feedstock collection, as current amount of poultry and livestock farming is increasing and manure management strategies are varied.

.....advanced algorithms and data analysis techniques are efficient pathways to create more positive impacts on quantifying biogas production variance (time requirement) and optimizing operation strategies to form beneficial products in realistic cases. A specific feedstock collecting plan should be identified, as the biogas supply on demand is significantly correlated with time dependency and transportation limits of organic waste management.

National deployment to meet the Chinese 1.5°C warming target

.... quantifying the characteristics and recovery percentages of different manure sources are still important task, due to the large spatial and temporal heterogeneity in estimating the associated biogas production potential. Furthermore, it also should be noted that long distances of feedstock transportation could possibly decrease the carbon mitigation contribution, owing to a large amount of fossil fuel use. The last but not the least, feedstock logistics would also be influenced by stakeholders’ revenues, transportation costs, and public interest for various rural communities, because the feedstock supply chain is a market interpretation. Thus, it is highly essential to identify the region- and practice-specific feedstock collecting strategy before the on-site CBPD deployments, including exploring specific biogas production rate models of segregated feedstock for co-fermentation with other kinds of agriculture waste, and properly organizing feedstock logistics to attain full-size biogas production. In addition, it is also important to clarify the coordinated scenarios for different feedstocks feeding.

Discussion

Such policy options must be evaluated in more detail to convince investors that upgrading CBPD is a reliable technology, as well as increase targeted subsidies for a combination of storage capacity optimization and operational training.

– Comment 10:

The figures and tables provided are relevant and support the text well, but they could be more effectively integrated into the narrative to enhance the reader's understanding.

Response: (marked in green in the revised manuscript)

- Thank you so much for your valuable comment. In the manuscript, the figures' presentation have been modulated from 4 pictures to 5 pictures, as schematic diagram of upgraded CBPD design (Fig. 1), curves fitting for GHG emissions analysis under different scenarios (Fig. 2), slipping avoiding and synergistic adjustment (Fig. 3), carbon mitigation of upgraded CBPD for universal deployment (Fig. 4), and manure availability and its further exploring potential (Fig. 5). Among, Fig. 3 is new to illustrate the most conflicted scenarios and its solutions, particularly for the coordinating mechanism to add an allowable fluctuation both in plant- or user- sides. Fig. 4 is refined to show the influence of solar-air temperature and the state-of-the-art decarbonization contributions on a world map. The original Fig. 5b has been separated into modified Fig. 5b and Fig. 5c, the modified Fig. 5b is used to express the performance of different CBPDs and illustrate the advantage of upgraded CBPD; Fig. 5c is used to illustrate the prospect of further carbon mitigation potential under different biogas use scenarios. Meanwhile, the Supplementary tables and Supplementary notes have been revised accordingly. Please check.

– Comment 11:

The references are comprehensive, but there is room for including more recent studies to strengthen the manuscript's grounding in the current state of research in this field

Response: (marked in green in the revised manuscript)

- Thank you very much for your valuable suggestion. It is been revised as instructed both in “References” of manuscripts and “Supplementary references” of Supplementary information, new studies are listed as follows. Please check.

References

9. Li, F., Zhang, J. & Li, X. Energy security dilemma and energy transition policy in the context of climate change: A perspective from China. *Energy Policy* **181**, 113624 (2023).
10. NDRC & MOA. The 13th Five-Year Plan for biogas Development of the People's Republic of China. (NDRC & MOA, 2017).
15. Sharma, V., Sharma, D., Tsai, M.-L., Ortizo, R. G. G., Yadav, A., Nargotra, P., Chen, C.-W, Sun, P.-P. & Dong, C.-D. Insights into the recent advances of agro-industrial waste valorization for sustainable biogas production. *Bioresource Technology* **390**, 129829 (2023).
16. EBA. Design, build, and monitor biogas and biomethane plants to slash methane emissions. (EBA, 2023).
18. Bakkaloglu, S., Cooper, J. & Hawkes, A. Methane emissions along biomethane and biogas supply chains are underestimated. *One Earth* **5**, 724–736 (2022).
29. Erickson, E. D., Tominac, P. A. & Zavala, V. M. Biogas production in United States dairy farms incentivized by electricity policy changes. *Nature Sustainability* **6**, 438-446 (2023).
31. Hoy, Z. X., Woon, K. S., Chin, W. C., Van Fan, Y. & Yoo, S. J. Curbing global solid waste emissions toward net-zero warming futures. *Science* **382**, 797-800 (2023).
32. Li, Y., Yan, B., Qin, Y., Shi, W. & Yan, J. Analysis of the types of animal husbandry and planting that influence household biogas in rural China. *Journal of Cleaner Production* **332**, 130025 (2022).
33. Gonzalez, R., Garcia-Cascallana, J. & Gomez, X. Energetic valorization of biogas. A comparison between centralized and decentralized approach. *Renewable Energy* **215**, 119013 (2023).
34. Svobodova, K., Owen, J. R., Kemp, Deanna., Moudrý, V., Lèbre, É., Stringer, M. & Sovacool, B. K. Decarbonization, population disruption and resource inventories in the global energy transition. *Nature Communications* **13**, 7674 (2022).
35. Emebu, S., Pecha, J. & Janáčová, D. Review on anaerobic digestion models: Model classification & elaboration of process phenomena. *Renewable & Sustainable Energy Reviews* **160**, 112288 (2022).
47. Yun, X. Shen, G., Shen, H., Meng, W., Chen, Y., Xu, H., Ren, Y., Zhong, Q., Du, W., Ma, J., Cheng, H., Wang, X., Liu, J., Wang, X., Li, B., Hu, J., Wan, Y. & Tao S. Residential solid fuel emissions contribute significantly to air pollution and associated health impacts in China. *Science Advances* **6**, eaba7621 (2020).
48. Fan, S. S., Jiang, M. H., Sun, D. Y. & Zhang, S. K. Does financial development matter the accomplishment of rural revitalization? Evidence from China. *International Review of Economics & Finance* **88**, 620-633 (2023).
51. Gao, Y. , Chen, M., Wu, Z., Yao, L., Tong, Z., Zhang, S., Gu Y. A. & Lou L. A miniaturized transit-time ultrasonic flowmeter based on ScAlN piezoelectric micromachined ultrasonic transducers for small-diameter applications. *Microsystems & Nanoengineering* **9**, 49 (2023).
52. Izadmehr, M., Shams, R. & Ghazanfari, M. H. New correlations for predicting pure and impure natural gas viscosity. *Journal of Natural Gas Science and Engineering* **30**, 364-378 (2016).
54. Li, J., Chen, C. & Liu, H. Transition from non-commercial to commercial energy in rural China: Insights from the accessibility and affordability. *Energy Policy* **127**, 392-403 (2019).
55. Liu, J., Zhou, X., Wu, J., Gao, W. & Qian, X. Heat transfer analysis of cylindrical anaerobic reactors with different sizes: a heat transfer model. *Environmental Science and Pollution Research* **24**, 23508-23517 (2017).

Supplementary references

3. Li, Y., Yan, B., Qin, Y., Shi, W. & Yan, J. Analysis of the types of animal husbandry and planting that influence household biogas in rural China. *Journal of Cleaner Production* **332**, 130025 (2022).
4. Bluemling, B. & Visser I. D. Overcoming the “club dilemma” of village-scale bioenergy projects—The case of India. *Energy Policy* **63**, 18-25 (2013).
5. Bakkaloglu, S., Cooper, J. & Hawkes, A. Methane emissions along biomethane and biogas supply chains are underestimated. *One Earth* **5**, 724-736 (2022).
6. Wechselberger, V., Reinelt, T., Yngvesson, J., Scharfy, D., Scheutz, C., Huber-Humer, M. & Hrad, M. Methane losses from different biogas plant technologies. *Waste Management* **157**, 110-120 (2023).

Reviewers' Comments:

Reviewer #1:

Remarks to the Author:

All my concerns and comments were well addressed. I don't have any further concerns or comments. I personally think this MS could be accepted for publication in Nature Communications.

Reviewer #2:

Remarks to the Author:

The authors have satisfactorily addressed my previous comments and suggestions, significantly improving the manuscript quality. However, there remains a persistent issue with using both the English language and technical terminologies throughout the manuscript. Therefore, before its possible publication in Nature Communications, the manuscript should be comprehensively revised focusing on refining the English language usage and ensuring clarity in technical terminologies.

Reviewer #3:

Remarks to the Author:

The authors have comprehensively addressed all the queries raised by the reviewers. Consequently, it is my opinion that the manuscript is now in a form suitable for acceptance and publication as it currently stands.

Reviewers' comments:

Reviewer #1: (marked in red for color highlighting in the revised manuscript)

– Comment 1:

The authors have satisfactorily addressed my previous comments and suggestions, significantly improving the manuscript quality. However, there remains a persistent issue with using both the English language and technical terminologies throughout the manuscript. Therefore, before its possible publication in Nature Communications, the manuscript should be comprehensively revised focusing on refining the English language usage and ensuring clarity in technical terminologies.

Response: (marked in red in the revised manuscript)

- Thank you very much for your comment. We have checked and revised as instructed, we also invited the language polishing agency “International Science Editing (<http://www.internationalscienceediting.com>)” to help this manuscript edition again. Particularly, we have replaced some technical terminologies with universal and exact description, and revised some ambiguous sentences to more smooth and understandable content. Please check.